# Past, Present, and Future Variability of Atlantic Meridional Overturning Circulation in CMIP6 Ensembles

Arthur Coquereau[1], Florian Sévellec[1], Thierry Huck[1], Joël J.-M. Hirschi[2], and Quentin Jamet[3]

[1]Laboratoire d'Océanographie Physique et Spatiale, Univ Brest CNRS IRD Ifremer, Brest, France
[2]Marine Systems Modelling, National Oceanography Centre, Southampton, SO14 3ZH, UK
[3]SHOM, Service Hydrologique et Océanographique de la Marine, Brest, France

**Correspondence:** Arthur Coquereau (arthur.coquereau@univ-brest.fr)

**Abstract.** The Atlantic Meridional Overturning Circulation (AMOC) is a key component of the climate system, exhibiting strong variability across daily to millennial timescales and significantly influencing global climate. Sensitive to external conditions such as freshwater input, greenhouse gas concentrations, and aerosol forcing, important variations of the AMOC can be triggered by anthropogenic emissions. This study presents a comprehensive analysis of sources of AMOC variance in state-of-the-art climate ensemble models. By decomposing the effects of scenario, model, ensemble, and time variability, along with their interactions, through an Analysis of Variance (ANOVA) and by introducing a novel combination of the variance components based on physical considerations, we identify three distinct regimes of AMOC variability from 1850 to 2100. The first regime, spanning most of the historical period, is characterized by a relatively stable AMOC dominated by internal variability (i.e., ensemble spread). The second regime, initiated by AMOC decline at the end of the 20th century and lasting until mid-21st century, is governed by a transient increase of time variability. Notably, the direct effect of forcing scenario differences remains muted all along this regime, despite the start of emission-scenarios in 2015. The third regime, beginning around 2050, is marked by the emergence and rapid dominance of inter-scenario variability. Throughout the simulations, inter-model variability remains the primary source of uncertainty, influenced by aerosol forcing response, AMOC decline magnitude, and the physical variability. A key finding of this work is the evidence that internal variability decreases simultaneously with AMOC intensity and seems inversely proportional to emission-scenario intensity.

## 1 Introduction

The Earth's climate is a complex system made up of various intertwined components interacting with each other and varying over a wide range of temporal and spatial scales. Among the major components, the Atlantic Meridional Overturning Circulation (AMOC) plays a key role by controlling a substantial part of the poleward heat transport with major impacts on the surrounding regions (Srokosz et al., 2012). At 26°N, it transports about 1.2 PW (1 PW = $10^{15}$ W) representing 60% of the net poleward heat flux of the ocean and 30% considering both ocean and atmosphere (Ganachaud and Wunsch, 2000; Trenberth and Fasullo, 2017; Johns et al., 2023). Several remote influences have also been identified, with the North East Brazilian and Sahel rainfalls, or the Atlantic hurricanes activity to only name a few (Knight et al., 2006). Finally, the AMOC contributes significantly to carbon sequestration through deep-water formation at high latitudes (Zickfeld et al., 2008).

Considering this leading role for the climate system, it appears critical to understand how the AMOC varies over time and identify the associated mechanisms. In this regard, a recent review provided a comprehensive description of the scales involved in AMOC variability (Hirschi et al., 2020). It shows that the variability unfolds over a large collection of temporal and spatial scales, driven by diverse physical processes—from daily to interannual and even decadal fluctuations. These include fast synoptic weather systems and near-inertial gravity waves, mesoscale eddies and large-scale baroclinic waves, as well as slower climate modes such as the North Atlantic Oscillation or the Atlantic Multi-decadal Oscillation. Accordingly, a common simplified distinction is made between intra-annual variability, primarily linked to Ekman wind forcing, and interannual-to-decadal variability, which is more strongly associated with geostrophic processes and large-scale density contrasts (see Buckley and Marshall, 2016, for a review). At the end of the spectrum, the AMOC has also been associated with abrupt climate events, such as Dansgaard-Oeschger and Heinrich events at millennial scale (Dansgaard et al., 1993; McManus et al., 2004; Böhm et al., 2015; Henry et al., 2016). These events have been associated with another important characteristic of the AMOC, namely its multi-stability (e.g., Sévellec and Fedorov, 2014). Indeed, a long literature has studied this aspect and shown that the AMOC is subject to collapses during which the system shifts from a first state with an intense circulation to another one where the circulation considerably weakens (Stommel, 1961; Lenton et al., 2008; Armstrong McKay et al., 2022).

Beyond these natural, internal, and chaotic variations of the AMOC, the system is also sensitive to external forcings, such as the anthropogenic emissions of aerosols and carbon in the atmosphere. These emissions can either reduce the surface temperature by increasing the reflection of incoming solar radiation (for the aerosols), or on the contrary, increase the temperature by intensifying the greenhouse effect (in the case of carbon dioxide or methane). In Subpolar North Atlantic regions, these changes in surface temperatures can have a direct impact on the buoyancy of surface waters, and can therefore reduce or intensify the formation of deep-water that need sufficient density to sink. In addition, human activities can impact the freshwater input in the subpolar gyre, for instance by modifying precipitation regimes or by boosting the Greenland Ice Sheet melting rate, and thus further increase the surface buoyancy and decrease dense water formation consequently (Gierz et al., 2015).

For our understanding of the future evolution of the AMOC and the potential impacts on the climate, it therefore appears crucial to study the different sources of variability and their role in the past, present, and future state of the AMOC. Our objective, in this work, is precisely to tackle this challenge and provide a comprehensive analysis of the AMOC variability in state-of-the-art climate models from the 6[th] Phase of the Coupled Model Intercomparison Project (CMIP6). In particular, we aim to separate the internal signal induced by internal modes of variability from the forced signal associated with anthropogenic fingerprint. The analysis benefits from a large body of work, over the last decades, dedicated to partitioning the variability (or the uncertainty) in climate simulations (Hawkins and Sutton, 2009, 2011; Yip et al., 2011; Sévellec and Sinha, 2018; Lehner et al., 2020; Zhang et al., 2023).

In this study, we take advantage of an improvement over the last generation of CMIP models, namely the presence of relatively large ensemble simulations. Each ensemble consists of several simulations of a single model covering the same time period with the same forcing, but starting with different initial conditions. This initial difference together with the chaotic nature of the system, then, allows the simulations to sample the internal variability by covering the phase space of the system. Traditionally, internal variability was assessed by computing the variance of the residuals of a climate variable (after removing

trends) over a fixed period, and was therefore often assumed to be stationary (e.g., Hawkins and Sutton, 2009). However, this hypothesis had already been highlighted in the seminal article (Hawkins and Sutton, 2009), and after being challenged in numerous articles, the latter demonstrated that the climate phase space evolves over time and that internal variability is anything but stationary when a forcing is applied (e.g., Cheng et al., 2016; MacMartin et al., 2016; Coquereau et al., 2025). It is therefore necessary to detach ourselves from this temporal dimension in order to accurately estimate this internal variability and its

evolution. To evaluate the role of forcing in climate variability, CMIP6 simulations provide different Shared Socio-Economic Pathways (SSP) allowing to investigate the response of the system under various forcing intensities (O'Neill et al., 2016). These scenarios extend after the 1850-2015 historical period up to 2100 (at least). The current analysis is based on 10 ensembles, each of them being produced with a different model, to draw a robust "model-independent" picture of the variability, and to investigate the sources of uncertainty associated with the representation of the AMOC in the different models.

To analyze this high-dimensional dataset and separate the different factors of variability, we used a proven and state-of-the-art method for climate datasets called Analysis of Variance (ANOVA, Zwiers, 1996; Wang and Zwiers, 1999; Hingray et al., 2007; Yip et al., 2011; Zhang et al., 2023). This method allows us to separate the total variability into the direct effect of each dimension (referred to as the "main effect" in the ANOVA framework) and the interactions among them. The ANOVA approach also enables us to move beyond another key assumption of the widely-accepted methodology (Hawkins and Sutton,

2009, 2011; Lehner et al., 2020)—namely, the additivity of variability/uncertainty components—by explicitly exploring their interactions. Zhang et al. (2023) computed an ANOVA decomposition involving three dimensions (3-way; i.e., models, real-izations, and scenarios) and focusing on ensembles. They applied this decomposition to temperature and precipitation, with a separation between ensemble members, scenarios, and models and showed that interactions account for almost half of the variance in surface temperatures. In this study, we propose incorporating the time dimension to examine how the interannual-

to-decadal variability of the AMOC—including long-term trends—has changed over time, by analyzing successive 30-year climate periods. Previous climate studies using ANOVA did not include time because the method was mainly used to measure uncertainty. A trend or event that is common across all scenarios, models, and ensemble members does not contribute to un-certainty per se, but rather to variability. By explicitly including time, we generalize the ANOVA framework to study not just uncertainty, but how variability itself changes over time. Accounting for time variability is particularly critical in the historical

period where there are no forcing scenarios. In such cases, neglecting the time dimension prevents the detection of externally driven changes in the climate state, for example those associated with volcanic eruptions.

It should be noted, however, that despite continuous progress, models still exhibit significant biases in their representation of the AMOC. This stems in part from the fact that the AMOC arises from a complex interplay of processes, many of which involve small-scale dynamics—such as mesoscale and submesoscale eddies or narrow boundary currents—that remain

unresolved in most climate models (Hirschi et al., 2020; Jackson et al., 2020, 2023; Gou et al., 2024). As a result, many simulations produce an overturning circulation that is too shallow and a western boundary current that is overly strong, while underestimating AMOC variability on interannual to decadal timescales (Jackson et al., 2023; Gou et al., 2024). Substantial model uncertainty also persists in estimates of the AMOC strength at 26.5° N, and models differ markedly in their depiction of overturning within the western subpolar gyre, as well as the water exchanges with the Indo-Pacific Ocean (Weijer et al.,

2020; Jackson et al., 2023; Baker et al., 2023). Beyond mean-state biases, differences in properties such as Labrador Sea salinity—sensitive to model and resolution—further influence AMOC by modulating the intensity and variability of dense water formation and, consequently, the strength and variability of the overturning itself (Jackson et al., 2020, 2023). It is therefore important to keep in mind the biases of these models when interpreting the results.

In the next section, we will present the ANOVA methodology and the data used for this study. Section 3 will be dedicated to the results. We will start with a general overview of the evolution of AMOC and the dominant sources of variance (3.1). Then, we will focus on the evolution of the different physical components of this variance during three periods of the time series representing three different regimes (3.2). The final part of the results section concerns inter-model variability and associated uncertainty (3.3). Finally, in section 4, we will summarize and discuss the results in order to shed light on possible future evolutions of the AMOC, and to discuss their meaning from the point of view of uncertainty and predictability.

## 2  Materials and Methods

### 2.1  Material

The present work is based on state-of-the-art climate simulations from CMIP6. The AMOC intensity is derived from the maximum meridional overturning streamfunction in the Atlantic at 26°N. Here, we focus on initial-condition ensemble simulations to sample the phase-space and investigate the spread of the different possible trajectory as a proxy of internal variability (Fig. 1). Initial conditions are derived following a predefined strategy for the CMIP6 framework, starting with different years of the multi-secular preindustrial control run (known as piControl), which is run under fixed external forcing conditions from the year 1850 (Eyring et al., 2016). This initialization strategy does not impose a specific clock to individual ensemble members such that it is not possible to statistically distinguish one ensemble member from another (although they might have dynamical peculiarities with important implications, e.g. Hawkins et al., 2016). This is true within an ensemble, but is also true across models and scenarios as a result of their common initialization procedure. When performing model averaging in ANOVA, nothing imposes that member #00 of ensemble A must be averaged with member #00 rather than member #01 of ensemble B. This provides a first insight that averaging involved in ANOVA will strongly impact ensemble statistics, as will be shown later on. Among the models, three offers relatively important sizes (with 25, 30, and 40 members, see Tab. 1), and we will mostly focus our analysis on these models, which we refer to as the "large ensemble models". We, nonetheless, extend the analysis to seven smaller ensembles (3-6 members) to test the robustness of our results and improve the representation of inter-model variability. We refer to these as the "small ensemble models". When performing inter-model statistics, we adopt a model democracy approach (Knutti, 2010), assigning equal weights to each model. The time series are separated in two parts: an historical period from 1850 to 2014 where the forcing are based on observations and a projection period from 2015 to 2100. For the 21[st] century, three major SSP are studied to estimate the forcing-scenario variability: SSP1-2.6 ("Sustainability"), SSP2-4.5 ("Middle of the road") and SSP5-8.5 ("Fossil-fueled development"). In the future scenarios, volcanic eruptions, which constitute short-term external forcings common to all members and can induce substantial climate variations, are not included, thereby removing a

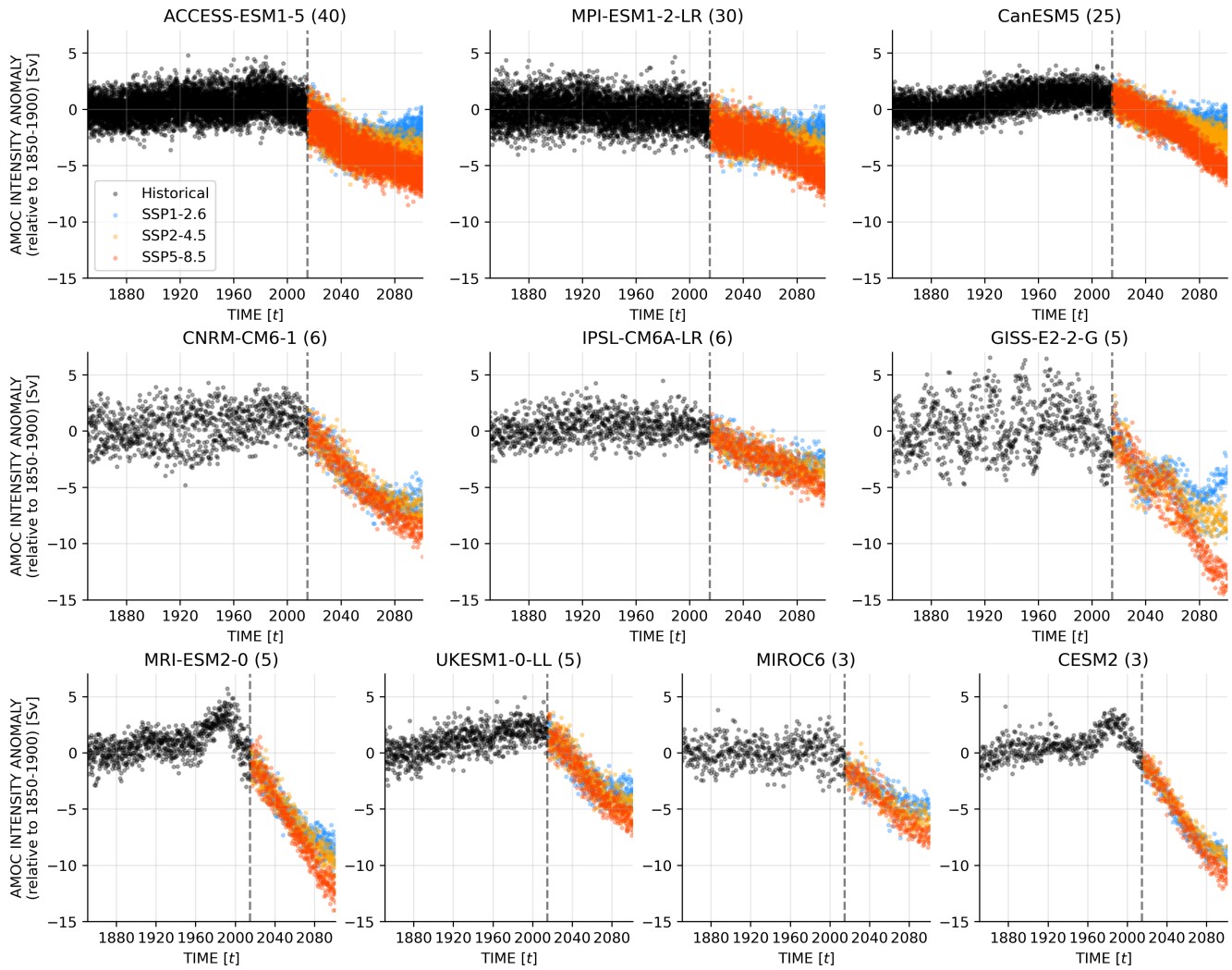

**Figure 1. Evolution of AMOC intensity anomaly.** Each subplot presents historical time series (black) followed by the three scenarios SSP1-2.6 (blue), SSP2-4.5 (orange), or SSP5-8.5 (red) for a given model. For each model, all realizations/members of the ensemble are shown, and the relative intensity uses as a reference the ensemble average between 1850 and 1900.

source of time variability. For more information on sources of climate variability in the SSP, see O'Neill et al. (2016). Overall, four dimensions are investigated: model, scenarios, ensemble, and time.

**Table 1.** List of models and members used in the study.

| Model | Ensemble size | Members | Reference |
|---|---|---|---|
| ACCESS-ESM1-5 | 40 | r[1-40]i1p1f1 | Ziehn et al. (2020) |
| MPI-ESM1-2-LR | 30 | r[1-30]i1p1f1 | Olonscheck et al. (2023) |
| CanESM5 | 25 | r[1-25]i1p2f1 | Swart et al. (2019) |
| MRI-ESM2-0 | 5 | r[1-5]i1p1f1 | Yukimoto et al. (2019) |
| MIROC6 | 3 | r[1-3]i1p1f1 | Tatebe et al. (2019) |
| CESM2 | 3 | r4i1p1f1, r10i1p1f1, r11i1p1f1 | Danabasoglu et al. (2020) |
| UKESM-0-LL | 5 | r1i1p1f1, r2i1p1f1, r3i1p1f1 r4i1p1f1, r8i1p1f1 | Sellar et al. (2019) |
| CNRM-CM6-1 | 6 | r[1-6]i1p1f1 | Voldoire et al. (2019) |
| IPSL-CM6A-LR | 6 | r1i1p1f1, r2i1p1f1, r3i1p1f1, r4i1p1f1, r6i1p1f1, r14i1p1f1 | Boucher et al. (2020) |
| GISS-E2-2-G | 5 | r[1-5]i1p3f1 | Rind et al. (2020) |

## 2.2 Methods

### 2.2.1 Analysis of Variance

The Analysis of Variance (ANOVA) method allows to decompose and attribute the variance in a multidimensional data set. It enables us to investigate the explanatory power of several qualitative factors (or dimensions, e.g. the various time steps, ensemble members, models, scenarios) on a quantitative variable (here, the AMOC intensity). It returns the role of each individual factor effect associated with single dimensions and the role of interactions associated with two or more dimensions. As described in Berrington de González and Cox (2007), interaction occurs when the separate effects of the factors do not combine additively. The ANOVA is thus generally represented as a linear decomposition of the total variance, with main effects depending on single dimensions and interactions depending on multiple factors that cannot be separated. The ANOVA is particularly useful when investigating three or more dimensions, which is difficult with more classical analyzes as covariance or correlation. There are, however, connections between covariance and ANOVA $2^{nd}$ order interactions. Both methods assess relationships between variables. The covariance measures how two variables change together, while the interaction in the ANOVA captures how a combination of two dimensions explains the variability in a dataset that would not be captured by either factor alone. Therefore, both refer to a joint variability.

Originally, the one-dimension ANOVA (referred to as 1-way ANOVA) was first employed by Yates (1938) in research on agriculture. Since then, it has been extended to the 2-way ANOVA and extensively used in climate science to characterize variability in simulations (e.g., Zwiers, 1996; Wang and Zwiers, 1999). In particular, different works used ANOVA to investigate the sources of uncertainty in climate projections, to separate model and scenario uncertainties in CMIP analyzes (Yip et al.,

2011), but also in more regional contexts (Hingray et al., 2007). More recently, a study took advantage of the larger ensembles available in CMIP6 to introduce the ensemble dimension directly in a 3-way ANOVA decomposition (Zhang et al., 2023). It therefore provides a separated dimension for internal variability, whereas this latter was previously represented as the residual
of the decomposition.

As explained in the introduction, in the present work, we build upon the previous 3-way ANOVA (Zhang et al., 2023) and extend it to a 4-way formulation by adding time dimension to investigate the changes of interannual-to-decadal time variability (i.e., dynamical adjustment). This is done by applying a 30-year rolling window (a typical climate period, discussed below) for the decomposition.

The main idea behind ANOVA is to separate the variance in a multidimensional data set by computing the variance after averaging the data over some of the dimensions. The importance of a given dimension is thus highlighted by the amount of variability removed when averaging this dimension. If a large fraction of the variability is removed therefore the dimension is an important factor of variability. At the origin, the multidimensional data set is not averaged and varies over the four dimensions $X(s,m,r,t)$, where $s$ represents scenario dimension, $m$ represents various models, $r$ the different realizations associated with
the ensemble dimension, and $t$ represents time. Then, the running window method is implemented by successively extracting 30-year subsets $x_\tau$ from the original dataset, each centered around a given time point $\tau$, as follows:

$$x_\tau(s,m,r,t) = X(s,m,r,t) \quad \text{for} \quad t \in [\tau - 15, \tau + 15]. \tag{1}$$

In the analysis, it is important to acknowledge the two time axes: $t$, on which statistical analyses are computed, and $\tau$, which is 30 years shorter and represents the centers of the successive windows of analysis. When averaging over a dimension, the data
set cannot vary in that dimension, this is signaled by the bar over the $x$ and by removing the dimension from the parentheses: for instance, $\overline{x_\tau}(s,r,t)$ represents an average among models, evaluated at time $\tau$ (See Section 2.2.3 for technical details). This central idea of ANOVA can be implemented for the four dimensions and lead to different remaining spread in the dataset after averaging some of the dimensions.

We will now use a few specific examples to detail the calculation of the main effect and interactions. The formulations can
be applied to all individual dimensions or combinations of dimensions. The ensemble main effect contribution ($V_r^{\text{main}}$ or called $R$ in the paper) is the variance of the dataset with respect to the ensemble of realizations, when averaging all dimensions but $r$. This corresponds to:

$$R(\tau) = V_r^{\text{main}}(\tau) = \frac{1}{N_r} \sum_{r=1}^{N_r} \left[\overline{x_\tau}(r) - \overline{x_\tau}\right]^2, \tag{2}$$

with $N_a$ the number of samples in dimension $a$, e.g. $N_r$ is the number of ensemble members. To compute the interactions,
we use the total variance budget involving several dimensions. The total variance involving ensemble and time ($r$ and $t$), for instance, corresponds to:

$$V_{rt}^{\text{total}}(\tau) = \frac{1}{N_r N_t} \sum_{r=1}^{N_r} \sum_{t=1}^{N_t} \left[\overline{x_\tau}(r,t) - \overline{x_\tau}\right]^2. \tag{3}$$

This total variance budget is also equal to the sum of the main effects of $r$ and $t$ and their interaction:

$$V_{rt}^{\text{total}}(\tau) = V_r^{\text{main}}(\tau) + V_t^{\text{main}}(\tau) + V_{rt}^{\text{interaction}}(\tau). \tag{4}$$

Thus, the 2$^{\text{nd}}$ order interaction between ensemble and time (called $RT$ in the paper) can be obtain with:

$$RT(\tau) = V_{rt}^{\text{interaction}}(\tau) = V_{rt}^{\text{total}}(\tau) - V_r^{\text{main}}(\tau) - V_t^{\text{main}}(\tau)$$

$$= \frac{1}{N_r N_t} \sum_{r=1}^{N_r} \sum_{t=1}^{N_t} \left[ \overline{x_\tau}(r,t)^2 - \overline{x_\tau}(r)^2 - \overline{x_\tau}(t)^2 - \overline{x_\tau}^2 - 2\overline{x_\tau} \left[ \overline{x_\tau}(r,t) - \overline{x_\tau}(r) - \overline{x_\tau}(t) \right] \right]. \tag{5}$$

We note that by averaging, the three terms in the last parenthesis will all become equal to $\overline{x_\tau}$. Thus, the expression can be simplified in:

$$RT(\tau) = \frac{1}{N_r N_t} \sum_{r=1}^{N_r} \sum_{t=1}^{N_t} \left[ \overline{x_\tau}(r,t)^2 - \overline{x_\tau}(r)^2 - \overline{x_\tau}(t)^2 + \overline{x_\tau}^2 \right]. \tag{6}$$

The more classical 2$^{\text{nd}}$ order interaction formulation of ANOVA (presented in (7) or in Zhang et al., 2023) is equal to (6), as the cross terms disappear when averaging:

$$RT(\tau) = \frac{1}{N_r N_t} \sum_{r=1}^{N_r} \sum_{t=1}^{N_t} \left[ \overline{x_\tau}(r,t) - \overline{x_\tau}(r) - \overline{x_\tau}(t) + \overline{x_\tau} \right]^2. \tag{7}$$

In the analysis, these interactions are named after the initial of the involved dimensions (e.g., $RT$ corresponds to the interaction between time and ensemble dimensions). For higher-order interactions, we use the same principle by removing lower order

terms from the total variance budget. For 3$^{\text{rd}}$ order interactions, $SRT$, for example, reads:

$$SRT = V_{srt}^{\text{interaction}} = V_{srt}^{\text{total}} - V_{sr}^{\text{interaction}} - V_{st}^{\text{interaction}} - V_{rt}^{\text{interaction}}$$

$$- V_s^{\text{main}} - V_r^{\text{main}} - V_t^{\text{main}}, \tag{8}$$

and for 4$^{\text{th}}$ order, $SMRT$, for example, reads:

$$V_{smrt}^{\text{interaction}} = V_{smrt}^{\text{total}} - V_{smr}^{\text{interaction}} - V_{smt}^{\text{interaction}} - V_{srt}^{\text{interaction}}$$

$$- V_{mrt}^{\text{interaction}} - V_{sm}^{\text{interaction}} - V_{sr}^{\text{interaction}} - V_{mr}^{\text{interaction}}$$

$$- V_{st}^{\text{interaction}} - V_{mt}^{\text{interaction}} - V_{rt}^{\text{interaction}}$$

$$- V_s^{\text{main}} - V_m^{\text{main}} - V_r^{\text{main}} - V_t^{\text{main}}. \tag{9}$$

    Other methods have also been used to investigate the different sources of variability/uncertainty in climate simulations, such

as the one used in Hawkins and Sutton (2009, 2011) or Lehner et al. (2020). While this method has the advantage of incorporating more models as it does not require ensembles, the internal variability is estimated as the residual of a polynomial fit. Here, given the importance of internal variability for our study, we decided to focus on ensemble simulations and use ANOVA. Another difference is the fact that the method developed by Hawkins and Sutton (2009) does not evaluate the importance of

interactions among sources of variance. While this simplifies the analysis, here we will see that interactions play an important role and must be considered to fully understand the evolution of AMOC variability.

Our results do not appear sensitive, especially qualitatively, to the size of the time window (Fig. B2). However, it should be noted that increasing the window size reinforces the main effect of temporal variance and reduces the ensemble variance main effect. The reason for this is explained in the next section. As 30 years is the typical period for studying the state of the climate (Arguez and Vose, 2011), we chose this duration to ensure a minimum coherence of the climate state and provide a good representation of temporal variance, while avoiding oversmoothing that would make transitions between regimes much more difficult to detect. Conversely, results concerning inter-model variability are sensitive to the AMOC reference chosen, particularly depending on whether we use AMOC anomalies or absolute intensities. This sensitivity is due to the persistent AMOC bias in climate models and the difficulty of capturing the right AMOC intensity (Weijer et al., 2020). In our work, we consider the AMOC anomaly relative to the period 1850-1900 because we are interested in the model uncertainty/variability associated with the model response to historical and future emissions rather than the initial and overall AMOC differences between climate models. Results with absolute AMOC values are discussed in section 3.3 and shown in Fig. B3.

### 2.2.2 Interpretation and combination of ANOVA components

The important number of separated components returned by the ANOVA decomposition requires an integrative approach if one wants to provide a conceptually coherent dynamical interpretation. Comparing the ANOVA with more standard method for studying the variability allows us to contextualize the meaning of each main effect and interaction. Sources of variability in a dataset are often characterized by computing the variance across the dimension of interest and averaging the results among other dimensions. An interesting property of the ANOVA is that this statistic can be directly retrieved by summing the main effects and the interactions associated with the given dimension (Fig. B1). As an example, ensemble studies on internal variability usually rely on the computation of ensemble variance (e.g., Coquereau et al., 2024), which can be reconstructed from the ANOVA with:

$$\text{Mean Ensemble Variance} = R + RT + SR + MR + SRT + SMR + MRT + SMRT. \tag{10}$$

This combination of components—referred to as variance reconstruction in our work—directly equals to the classical variance, allows to better understand the ANOVA decomposition. However, with this reconstruction, the interactions are embedded within each interacting dimension and, thus, would be accounted multiple times when deriving all dimensions. For example, $RT$ will be taken into account both in the mean variance of the ensemble and in the mean variance of the time. As a result, the sum of the variance components would exceed the total variability of the dataset, i.e.:

$$\text{Total Variance} \leq \text{Mean Ensemble Variance} + \text{Mean Time Variance}$$
$$+ \text{Mean Model Variance} + \text{Mean Scenario Variance}. \tag{11}$$

An alternative combination method have been proposed to provide an overall picture of the sources of variability without over-estimating the total variability. Zhang et al. (2023) proposed a statistical separation of the sources based on the division of

interactions between the involved components. For example, the $2^{nd}$ order interaction between time and ensemble dimensions (i.e., $RT$) is divided by two and each half is allocated to one dimension: time or ensemble. Following this statistical separation method, the calculation for ensemble dimension leads to:

$$\text{ENSEMBLE} = R + \frac{1}{2}(RT + SR + MR) + \frac{1}{3}(SRT + SMR + MRT) + \frac{1}{4}SMRT. \tag{12}$$

In the next section, we will employ this separation method to provide an overview of the variance distribution. However, this method does not allow for a direct comparison with traditional variance computation (e.g., Mean Ensemble Variance). Moreover, the statistical separation relies on a dimensional-wise approach rather than physical considerations. As a result, a single physical mechanism driving changes in variance may influence several variance components simultaneously. For instance, the $ST$ components, which might highlight differences in trends between scenarios and therefore appear naturally linked to the scenario dimension, would nevertheless be partitioned equally into the time dimension. Finally, compensations can occur between ANOVA components and cause spurious results with the statistical separation method.

Beyond these combination methods (i.e., variance reconstruction or statistical separation), in the present study, we will analyze each component individually, as individual components provide important insights on the evolution of variance in the dataset.

Main effects are the most intuitive components of the ANOVA, as they represent the variance across a single dimension when other dimensions are averaged. As an illustration, model main effect ($M$) corresponds to the differences between model averages, where the mean across ensemble, scenarios, and time is computed for each model. The interaction terms can be more challenging to interpret as they represent the variability located in multiple dimensions. Approaching them from the perspective of internal variability makes it easier to understand their physical significance as this variability represent the full variability of the physical system under steady external forcing conditions (considering a single ensemble model). Consistently with the variance reconstruction method (10), the mean ensemble variance is exactly equal to the sum of the ensemble main effect ($R$) and all interactions involving the ensemble dimension. This highlights that each dimension includes a part of internal variability. For time and ensemble dimensions, this is somehow natural, but it is also true for scenarios. When the trajectories separate under different forcing scenarios, a slight phase-shift of internal variability appears. This phase shift represents the fraction of internal variability associated with the scenario dimension. Averaging over scenarios removes part of this phase shift and, consequently, part of the internal variability, similarly to averaging over a time window or across realizations. If all dimensions are sufficiently large (many members, many scenarios, long time period), the interaction between scenarios, realizations, and time ($SRT$) should capture the entire internal variability. However, if the dimensions are too small, $SRT$ will not capture this entire variability, and the importance of $SRT$ decreases, compensated by an increase in other main effects or interactions. To illustrate this, we designed, and analyzed with ANOVA, a synthetic model providing AMOC time series that closely mimic CMIP6 simulations (see Section A). This especially allows us to modulate the number of scenarios, which is challenging with "real" AMOC simulations. Increasing the variety of scenarios, lead to an increase of $SRT$ and a decrease of $RT$, which demonstrate the presence of internal variability in the scenario dimensions as explained previously (Fig. A1). This corresponds to a "relocation" (as used later) of variability from one component to another—i.e., a shift in the component where

variability is detected by the ANOVA. If the number of scenario is too small, only a part of internal variability will be canceled by averaging scenarios.

The same logic applies for time dimension. During the CMIP6 historical period, $R$ represents the part of the internal variability that is not captured in $RT$, i.e., the internal variability with a period larger than the rolling time window of 30 yr. The sensitivity test on time windows depicts an anti-correlation of $R$ with $RT$, and with the length of the time windows (Fig. B2). Therefore, $R$ represents the low-frequency internal variability, and when the size of the time window increases there is, logically, less variability at lower frequency. After the beginning of scenarios this low-frequency internal variability shifts from $R$ to $SR$, if the number and variety of scenario are sufficient. The fact that all components involving $S$ are null at the beginning of the time series is due to the fact that all scenarios are merged, making $SR$ and $SRT$ equal to zero because averaging over scenarios does not remove any internal variability. Finally, $ST$ represents the part of the scenario dispersion removed by the inherent smoothing of the time averaging, as shown in the sensitivity test on time window size (Fig. B2), where increasing the window size decreases $S$ and increases $ST$, and vice versa.

### 2.2.3 Bootstrapping

To obtain robust results and assess the model uncertainty it is important to mix the different models. As the ensemble models have different sizes, we used a Bootstrap methodology to aggregate them. Only models of comparable size are combined. Large ensemble models (with 25 to 40 members) are mixed together and smaller ensemble models (from 3 to 6 members) are mixed together. The bootstrapping procedure is relatively simple. For each large ensemble model, we select randomly 20 members (with replacement) which are assembled with 20 members of each other large ensemble model. The ANOVA decomposition is thus applied on the 60 selected members from the three models. The selection of 20 random members and the ANOVA decomposition is then replicated a given number of time. In our case we use 100 resamplings and we did not observe substantial improvements by further increasing this number (e.g. to 1000). The results of the 100 resamplings are then averaged and analyzed. For the smaller members, the procedure is similar but by selecting 3 members per model instead of 20.

A sensitivity test was carried out to assess the impact of sub-sample size for large ensemble models (Fig. B4a, c, e and B5a, c). The test indicates that reducing the subsample size from 20 to 3 members leads to a decrease of internal variability and an increase of time variability. Specifically, during the historical period, we observe a transfer from $R$ and $RT$ to $T$, and a transfer from $RT$ and $SRT$ to $ST$ in the 21$^{\text{st}}$ century. The size of the ensemble has therefore a direct impact on the level of internal variability. This is because when three ensemble members are used, the ensemble average does not completely eliminate internal variability. We also assessed the impact of a mixture of large and small ensemble models versus the use of small ensemble models only. In this case, we applied 3-member resampling for all ensemble models. While the model-associated factors of variability are unaffected (Fig. B5), the mixture leads to a decrease in all physical factors. The decrease is relatively homogeneous between components, as the variance distribution remains unchanged (Fig. B4f).

## 3 Results

Before delving into the various factors and contributions of the variance, we will start by drawing a general picture of the AMOC intensity evolution in the CMIP6 dataset to better understand the changes of variability.

The ensemble-averaged AMOC time series for different models and scenarios appear relatively stable during most of the historical period, from the mid-19[th] to the mid-20[th] (Fig. 2a and b). Some models then present a weak increase often attributed in the literature to increasing aerosol concentration (Menary et al., 2013, 2020; Robson et al., 2022). In the last decades of the 20[th] century, the AMOC intensity initiates a substantial decrease, with no visible differences among emission pathways up to the middle of the 21[st] century. Afterward, the scenarios start to separate with the strongest forcing presenting a continuous decline, while the weakest forcing scenario stabilizes or even slowly recovers depending on the models. At the end of the projection period, while the large ensemble models seem to converge under a given scenario in terms of relative decrease, the small ensemble models present growing differences.

This brief overview reveals substantial AMOC variability across time, scenarios, and models—consistent with previous studies (e.g., Weijer et al., 2020; Jackson et al., 2023)—from the last decades of the 20[th] century. This behavior is expected to persist and even intensify throughout the 21[st] century.

### 3.1 A general picture of the variability

To analyze the evolution of the variability and its various contributions, we used the Analysis of Variance (ANOVA) method, described in section 2.2.1. The total variance from the multidimensional dataset is thus split into different contributions separated into main effects and associated interactions. Again, given the number of components, it is helpful to start the analysis simple and progressively increase complexity. For this purpose, we take advantage of the statistical separation method proposed by Zhang et al. (2023, and summarized in (12)). For each dimension of the dataset, we gather its main effect and redistribute, with equivalent weights, the interaction terms (Fig. 2c-f).

The inter-model variability (representing differences among models) appears to be the dominant factor of variance among most of the studied period. After a low-level of variability up to the beginning of the 20[th] century, due to the AMOC intensity reference taken as the 1850-1900 model average, the contribution rapidly increases to dominate both in large and small ensemble models. Two consecutive periods of increase detach from this time series. The first, which reached its peak in the second half of the 20[th] century, is likely associated with the differences of AMOC response to aerosol forcing among models (Menary et al., 2013, 2020; Robson et al., 2022). This increase is shifted in time between large and small ensemble models. While it starts around 1940 in the larger ensembles, it does not appear until 1970 in the smaller ones. The second increase, starting at the very end of the 20[th] century, seems due to the important AMOC decline and to the differences of decline magnitude among models. For example, it has been shown that the rate of AMOC weakening is linked to surface salinity and dense water formation in the Labrador sea and further influenced by model horizontal resolution (Jackson et al., 2020, 2023). Historical AMOC strengths and pathways observed in models also appear to correlate with their rate of weakening (Baker et al., 2023).

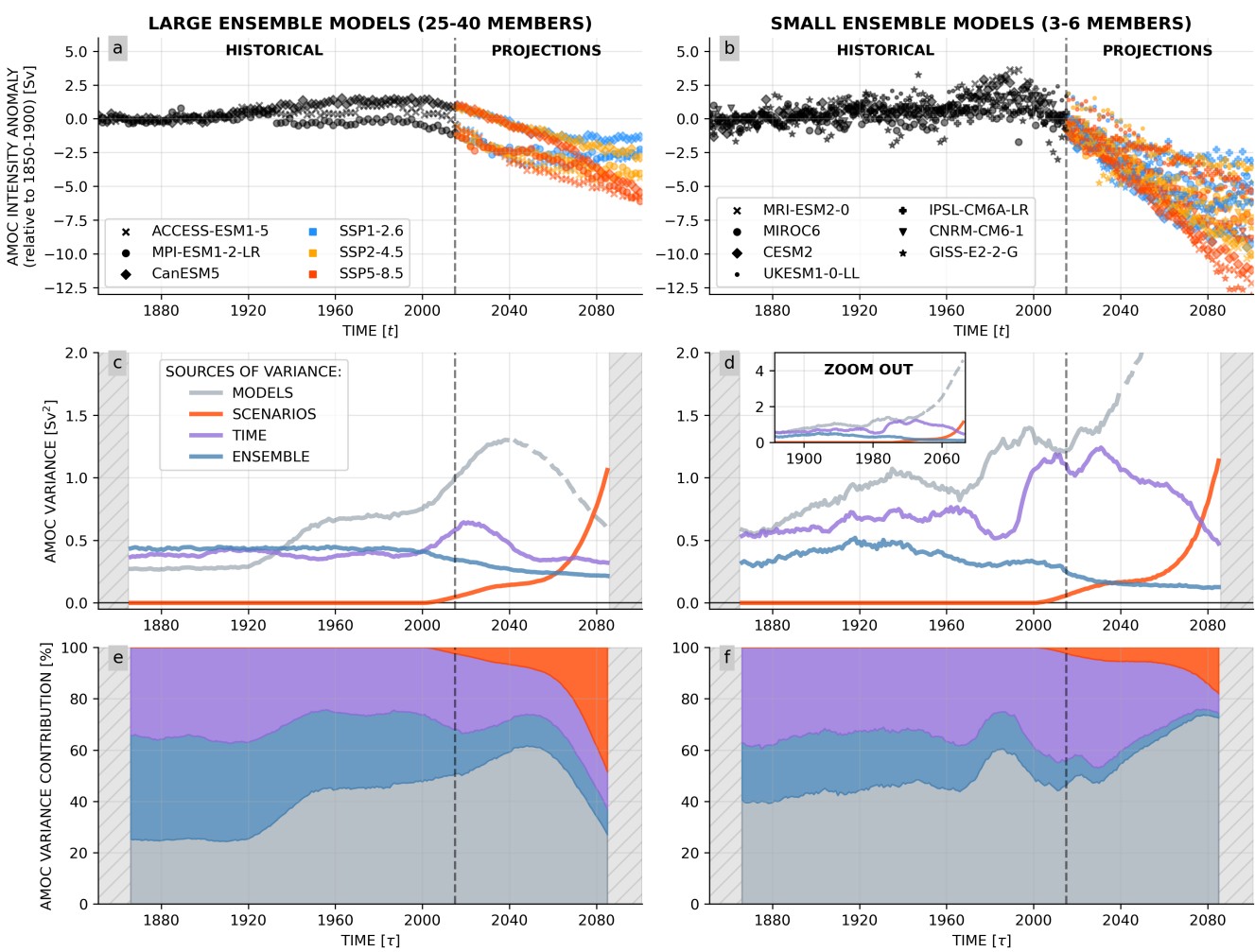

**Figure 2. General picture of the AMOC evolution at 26°N from CMIP6 ensemble models.** (a, b) Ensemble-averaged AMOC anomaly time series over historical (1850-2015, black) and projection (2015-2100, colors) periods. Reference value is the individual model average from 1850-1900. (c, d) AMOC variance associated with each of the four general factors based on the statistical separation methodology of Zhang et al. (2023). Model contribution has been represented with a dashed line after 2040 to highlight the lack of consistency between c and d regarding this result. Small and large ensemble models have the same y-axis, and a zoom-out box is displayed for the variance of the small ensemble models to show the full increase in model contribution (d). (e, f) Contribution of each factor to the total variance. Results are presented for large ensemble models (a, c, e) with 25-40 members and small ensemble models (b, d, f) with 3-6 members.

After 2040, the trajectory of the time series becomes increasingly uncertain, as evidenced by the growing divergence between small and large ensemble models (grey dashed line). This uncertainty partly arises from differences in AMOC sensitivity to external forcing between the two groups. Specifically, the three large ensemble models show similar AMOC sensitivity, while the seven small ensemble models exhibit greater diversity in their responses to external forcing (e.g., Fig. 2 in Weijer et al., 2020). This study, which focuses on AMOC anomalies relative to the historical state (defined as the 1850-1900 mean), emphasizes the AMOC's sensitivity to external forcing, i.e. its tendency to shift states as forcing deviates from the historical baseline. Consequently, the small ensemble models, representing a broader range of sensitivities, exhibit an increasingly larger inter-model spread over the second half of the 21$^{st}$ century. In addition to the distinct characteristics of the individual models within each group and the greater number of small ensemble models analyzed, this divergence is also potentially associated to the poorer separation of internal variability in small ensembles which may contribute to differences between models (Bonnet et al., 2021), although selecting only three members from each large ensemble model does not increase model variability (Fig. B5).

Ensemble and time variability are the two other important factors of variance during the historical period, with comparable orders of magnitude. Nevertheless, they have very different evolutions. Time variability is relatively stable during most of the historical period. Then, at the end of the 20$^{th}$ century, it presents a transient increase for a few decades as a signature of the AMOC decline. It is particularly striking in large ensemble models, where the time variance bump is perfectly phased with the AMOC decline and the associated second increase of inter-model variability. In small ensemble models, the transient aspect of the increase is slightly less clear, but still visible. The ensemble spread is also very stable at the beginning of the historical period. Then, concomitantly to the AMOC decline, it initiates a slow and progressive decrease up to the end of the 21$^{st}$ century.

Finally, the last component is the scenario variability, mainly associated with the divergence of future emission pathways. By definition, it is zero at the outset, since the scenarios only start in 2015. At the beginning of the 21$^{st}$ century (due to the 30-year running window), it thus starts to increase but stabilizes at a low level until the mid-21$^{st}$ century. After which, it rapidly increases to dominate over both internal and time variability in small ensemble models, and even over the inter-model variability in large ensemble models.

To move a step further in understanding these changes of variability and the various phenomena involved, we will go beyond the statistical separation and analyze each individual contribution, including the interactions. As we shall see, interaction terms can be responsible for a large part of the variability and provide crucial information that must be analyzed.

## 3.2 A focus on the three regimes of variability

In this section, we leave aside the model-associated factors of variance that do not refer to directly observable variability, and focus on analyzing the "physical factors" of variability. As it shows the robustness of the study, model-associated variability and interactions are discussed later in subsection 3.3. In that subsection, the analysis of the model-associated interactions shows that they are mainly governed by physical factors. This makes it more natural to analyze the physical factors first, even if their associated variance is smaller.

Three phases and regimes of variability can be isolated in the simulated time series from 1850 to 2100. Each of them will be analyzed in the following subsections.

Special attention is paid to the analysis of large ensemble models as they provide a better cover of the phase-space and thus a more accurate picture of the variability, especially associated with internal factors. However, the small ensemble models remain important both for assessing the consistency of the results and for the analysis of model-associated variability (since

there are more models with a small ensemble) performed in section 3.3.

### 3.2.1   First century of the historical period controlled by internal variability

Among the different physical factors (Fig. 3), the interaction between time and ensemble ($RT$) dominates the variability during most of the historical period: from the mid-19[th] to the last decades of the 20[th] century. It is particularly striking in large ensemble models where it represents around 80% of the physical factor variability (Fig. 3e). In small ensemble models, it is

lower but still represents approximately 40% of the total physical factor variability (Fig. 3f). The remainder is divided into the main effects of ensemble ($R$) and time ($T$).

In a relatively stable context, such as the beginning of the historical period, quasi-ergodicity can be assumed (Hingray and Saïd, 2014). Ergodicity defines a situation where the ensemble and time statistics tend toward the same values. Here, we cannot speak of ergodicity because of the impact of the trend on time variance. However, as the trend is small during this period so is

its impact on time variance allowing to use the term quasi-ergodicity. This can be highlighted by computing the ensemble-to-time variance ratio (Fig. 4a). During this regime, the ratio appears close to one, indicating relatively similar ensemble and time variance and thus quasi-ergodic conditions. In this context, internal variability is the dominant component among the "physical factors" of variability. This reflects in the $RT$ interaction that gathers most of the internal variability, since a large part of the variability is removed in this period if either time or realizations are averaged.

A small fraction of internal variability remains, however, present in the $R$ main effect and corresponds to the internal variability with period larger than the filtering window, as explained in section 2.2.2 and shown in Fig. B2, where the importance of $R$ appears anti-correlated with the size of the time window. This fraction is relatively similar between the two categories of models.

The fraction of variability located in $T$ originates from two sources: the weak non-ergodicity, and the limited ensemble size.

The weak non-ergodic aspect of the evolution is associated with the weak linear trend over this period and the response of the AMOC to aerosol forcing. This effect corresponds approximately to the magnitude of $T$ seen in large ensemble models. Alternatively, the size of ensembles is mostly observable in small ensemble models, where the number of realizations is too limited and do not allows to cover the entire phase space and completely remove the internal variability when averaging. Thus, a significant part of internal variability remains after averaging realizations as observed with the larger magnitude of $T$ in small

ensemble models. This impact of ensemble size is demonstrated by the sensitivity test on number of realizations in Fig. B4. This test shows that when decreasing the number of realizations taken from large ensemble models, a significant fraction of internal variability from $RT$ is relocated to $T$, leading to a repartition similar to small ensemble models.

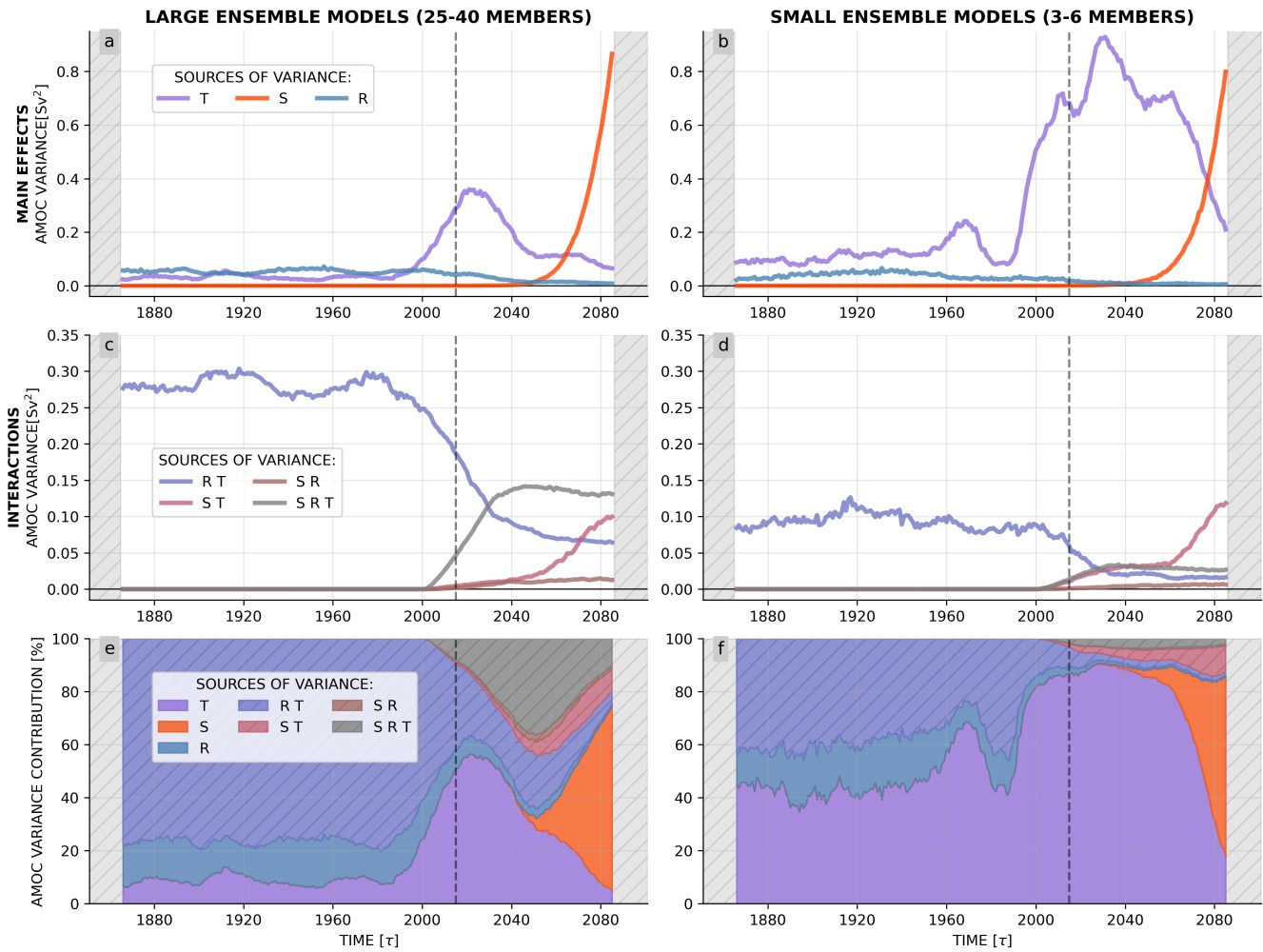

**Figure 3. Evolution of the variability of the physical factors including ensemble, time, and scenarios dimensions.** (a-d) Variance associated with each main effect (a, b) and their interaction (c, d). (e, f) Relative variance contribution of each main effect and their interactions to the total physical factor variability. Interactions are highlighted by hatches. Letters refer to the dimension involved in the variance calculation, with a single letter for main effects and a combination of letters for interactions: S refers to the scenario dimension, T to time, and R to realizations (see section 2.2.1). Results are presented for large ensemble models (a, c, e) with 25-40 members and small ensemble models (b, d, f) with 3-6 members.

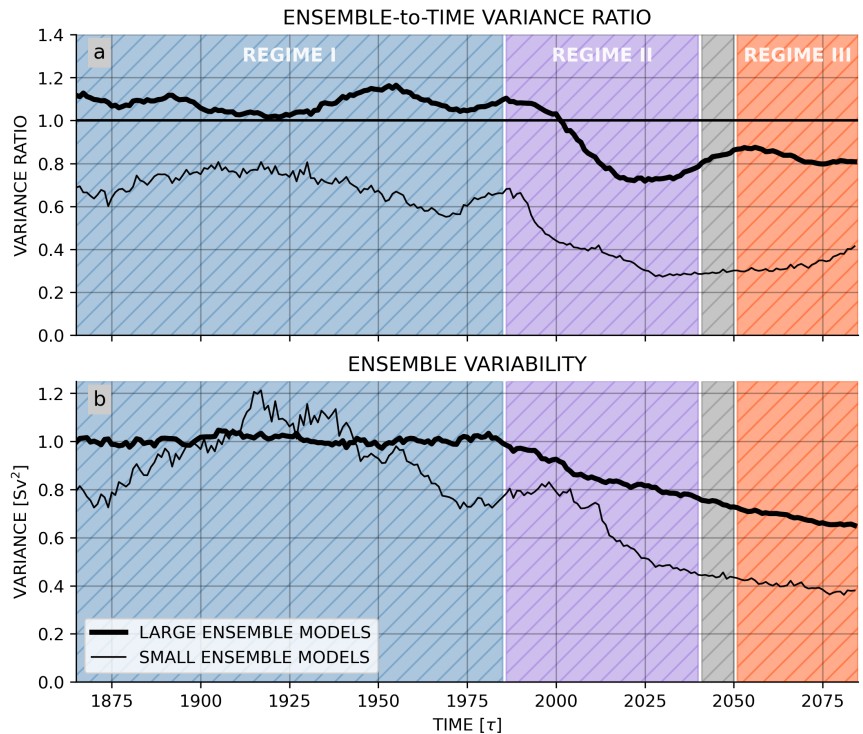

**Figure 4. Time and ensemble variance** (a) Comparison of their variability with the ensemble-to-time variance ratio. We use this ratio as a proxy of ergodicity. Here ensemble and time variance are derived from the sum of interactions according to the variance reconstruction method described in section 2.2.2. Similar results are obtained using the statistical separation methodology proposed by Zhang et al. (2023). (b) Evolution of ensemble variability (i.e., inter-member spread) over time. Thick lines represent large ensemble models and thin lines small ensemble models.

Going into detail, the ensemble-to-time variance ratio appears slightly smaller than one for small ensemble models, whereas it is slightly greater than one for the large ensembles (Fig. 4a). This can be directly linked to the previous results showing that the repartition of variance between $T$ and $RT$ components is sensitive to the ensemble size. The statistical separation method used in Fig. 4, attributes the variance in $RT$ to both ensemble and time dimensions, ant the variance in $T$ to time dimension only. Thus, when smaller ensemble models present a relocation from $RT$ to $T$, it results in a decrease of ensemble-to-time variance ratio. In a ideal case, with a sufficient large ensemble model, one would expect the ensemble variance to be slightly larger than the time variance, since it allows to capture all frequency of internal variability while time variance miss by definition the periods larger than the time window. This effect is highlighted by the long de-correlation timescale of AMOC trajectories in CMIP6 (calculated as the e-folding timescale of the auto-correlation) that appears to range between 2 and 40 years (Fig. B6) suggesting non-negligible low frequency variability in the data.

The decomposition of the total variance budget involving time and ensemble dimensions is particularly insightful to understand the role of these two dimensions (Fig. 5a and b). It corresponds to the sum of the main effects and their interaction,

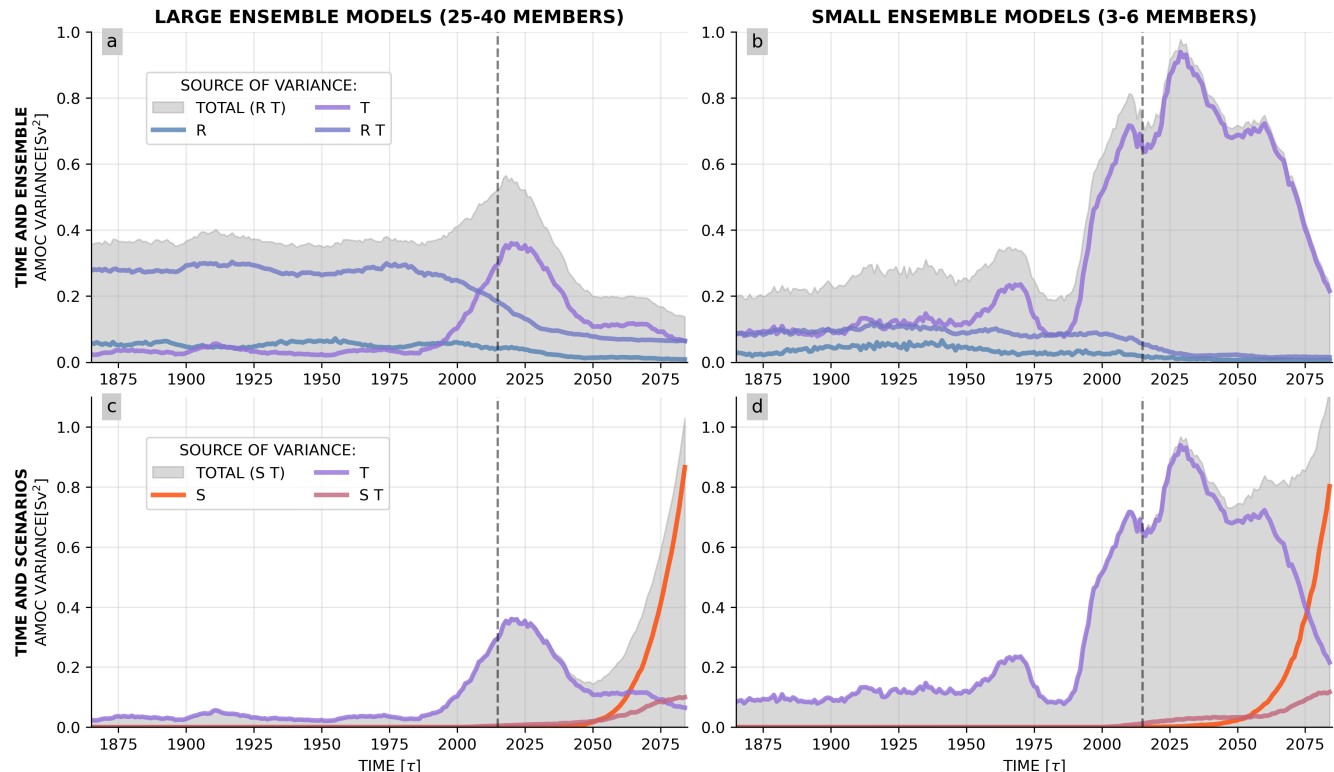

**Figure 5. Evolution of variance associated with time and ensemble (a, b) and time and scenarios (c, d).** Grey shaded areas represent the total variance associated with the considered dimensions. It corresponds to the sum of the colored lines representing the different variance contributions including the two main effects and their interaction. Letters refer to the dimension involved in the variance calculation, with a single letter for main effects and a combination of letters for interactions: S refers to the scenario dimension, T to time, and R to realizations (see section 2.2.1). Results are presented for large ensemble models (a and c) with 25-40 members and small ensemble models (b and d) with 3-6 members.

as shown in (4). Following (3), it is also equal to the variance of the initial dataset after averaging models and scenarios. The decomposition of the variance clearly highlights the dominant role of the interaction especially in large ensemble models (Fig. 5a). In small ensemble models, the interaction is much lower consistently with previous results on subsample size. The time main effect is larger than the ensemble main effect showing that quasi-ergodicity is not reached because of the too small ensemble size.

The first century of the historical period is therefore characterized by strong, stable internal variability, with a major role for the $RT$ interaction underlining the quasi-ergodicity of the regime. Scenario variability remains zero, by definition, since the scenarios only start in 2015.

### 3.2.2 AMOC decline insensitive to forcing-scenarios up to mid-21$^{st}$ century

After the first regime of relatively stable AMOC intensity and variance components, the AMOC enters a phase of intense
decline reflected both in the statistical separation (Fig. 2) and by the substantial and transient increase of $T$ main effect (Fig.
3e and f). This regime is thus associated with a clear loss of ergodicity (Fig. 4a), where the ensemble-to-time variance ratio
presents a decrease of 40 to 60 % for large and small ensemble models, respectively.

Around 2000, the increase in the time main effect caused it to exceed the ensemble main effect in the large ensemble models
(Fig. 5a). In addition to the increase of $T$, this change of dominant factor is also due to the decline of $R$, that follows the same
decreasing pathway as $RT$. During this regime $R$ losses around 3/4 of its variance in both ensembles (Fig. 3).

From 2015, scenarios start and the associated variability emerges (Fig. 2). This emergence is, however, not generated by the
scenario main effect ($S$) but by the interaction between time, scenarios, and ensemble ($SRT$, Fig. 3). This increase of scenario-
associated variability before stabilizing, is therefore not the direct effect of the forcing but, rather, a sort of chaotic internal
variability triggered by slight differences among scenarios allowing the simulations to spread. Indeed, during the historical
period, the three time series corresponding to scenarios for each member of models are perfect replicas of the historical
simulation. However, when the scenarios formally begin in 2015, small differences or perturbations in forcing emerge among
them due to the progressive separation of scenario trajectories—for instance in terms of $CO_2$ emissions (O'Neill et al., 2016).
These small differences are amplified by the chaotic nature of the system, causing the three scenario time series to spread
similarly to pseudo-ensemble members. This internal and chaotic nature of $SRT$ variability is underlined by the magnitude of
this variability, which appears to be three times as large in large ensemble models as in small ones, consistently with $RT$ that
is greater in large ensembles. The absence of direct scenario main effect variability is also evident by $S$, that remains at zero
before 2040-2050 (Fig. 5c and d). This particular behavior was highlighted by Weijer et al. (2020) and acknowledged in the
latest Intergovernmental Panel on Climate Change (IPCC) report (AR6, Lee et al., 2021, p. 576).

This internal and chaotic aspect of $SRT$ interaction provides a first explanation for the common decline of $R$ and $RT$, due
to the relocation of a substantial fraction of the internal variability originally located in these two components toward $SRT$
(consistently with interpretation of interactions provided in Section 2.2.2). However, this decrease may also be driven by an
overall decline of the total internal variability if the latter is sensitive to the AMOC intensity. We investigate this second factor
by computing for each individual model and scenario the evolution of ensemble variance over time (Fig. 6). This analysis
shows an overall decrease of the ensemble variance over time, illustrating a contraction of the phase-space. To assess this
visual decline, we compute the average ensemble variance over 50-year time window in the historical and projection periods.
In the historical period, we select the 1900-1950 window, located a few decades after the beginning of the historical simulations
(allowing for possible AMOC adjustment from the control preindustrial forcing to the historical one) and before the increase
of aerosol forcing. For the projection, we select the last 50 years to target the potential maximum impact of scenarios. In all
models, the 1900-1950 period presents a larger ensemble variance than the 2050-2100 one, with an intensifying decrease over
time. If we compare now the different scenarios, the majority of models (7/10), including the three large ensemble models,
presents weaker ensemble variance under SSP5-8.5 than under SSP1-2.6; 6/10 models present weaker variability under SSP5-

8.5 than SSP2-4.5; and 8/10 a weaker variability under SSP2-4.5 than under SSP1-2.6. For the three large ensemble models, the ensemble spread is sorted according to scenario intensity, except for CanESM5 where SSP2-4.5 shows a slightly weaker variance than SSP5-8.5. These results therefore suggest a strong link between forced AMOC weakening and decrease in the AMOC ensemble variance. The decline of $R$ and $RT$ seems, therefore, driven by the combined effect of a declining total internal variability and a relocation of a part of this variability from $R$ and $RT$ toward $SRT$ under the effect of the scenario-perturbed pseudo members.

Finally, the last component evolving during this period is the interaction between time and scenarios ($ST$) that represent in a way the scenario uncertainty on temporal trends. The very weak increase of $ST$ when scenarios appear, remaining at a very low level up to 2050, especially in large ensemble, underline the previously discussed absence of inter-scenario differences before mid-21st century. The analysis of terms included in the total variance involving time and scenario dimensions also confirms that time is by far the leading factor of the associated variability during this phase, with $S$ and $ST$ negligible compared to $T$ (Fig. 5c and d). During this regime, while the scenarios have started, the AMOC variability remain driven by the declining trend associated with a dynamical-adjustment without direct impact of forcing scenarios.

Beyond the AMOC decline associated with the transient increase in time variability, the $T$ main effect initially decreases and then stabilizes together with $SRT$ around 2050. $RT$ interaction decrease rate also flattens, while $ST$ only exhibits a very weak increase. Before the start of the third regime, we therefore observe an intermediate phase where all components of variability appear stable. This short phase around 2040-2050 is dominated by the $SRT$ interactions associated with a "scenario-perturbed" internal variability. This is therefore in a way another phase were internal variability is important despite the significant trend. While most of the components in small ensemble models present a clear signature of this intermediate regime, it is less clear for $T$ main effect. We suggest this is due to the less accurate diagnostic of internal variability in small ensemble models, hence impacting the separation of internal and dynamical-adjustment variability.

### 3.2.3  Late separation of anthropogenic emissions scenarios after the mid-21st century

The third regime starts with the emergence of the scenario main effect $S$. It presents a steep increase from the middle of the 21st century and becomes, in a few decades, the dominant factor of physical variability. In parallel, the $ST$ interaction presents a weaker but still substantial increase, since the scenarios have an important role in AMOC evolution over time. As described in Section 2.2.2, the $ST$ interaction also represent the smoothing of scenarios by the time window and can thus be directly associated to scenario variability and effect.

The scenario variability becomes the first order factor of variability even above inter-model variability in large ensemble models over the last decades of the 21st century. The increasing and prominent role of scenarios/forced variability demonstrates that the system leaves a regime driven by past, historical forcings to enter in a phase of forced evolution driven at first order by the anthropogenic emissions scenarios.

After focusing on the physical factors of variability, we can investigate variability associated with inter-model differences and the drivers of these differences throughout the multisecular simulations. This will also measure the confidence and robustness of our physical factor results.

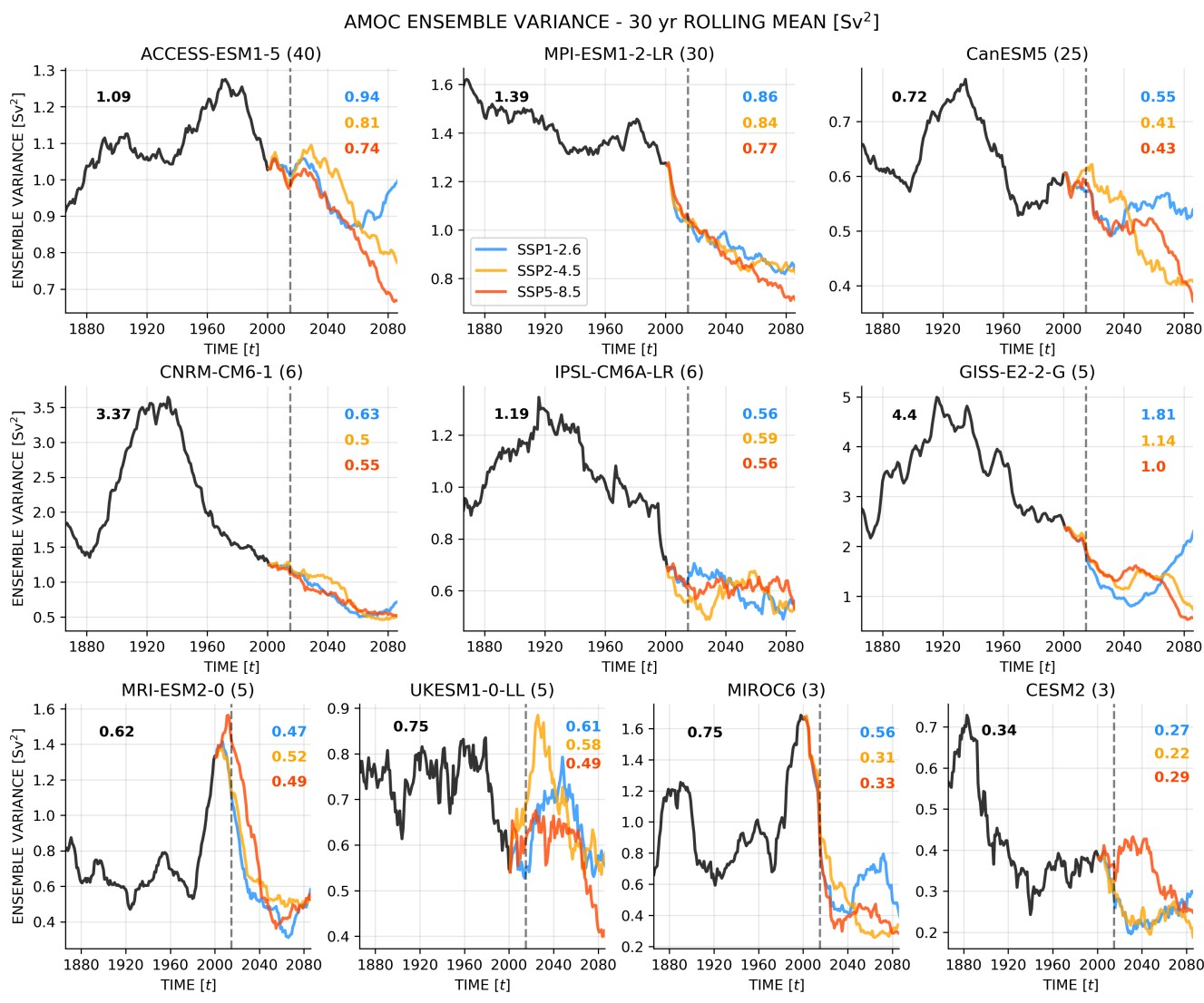

**Figure 6. Evolution of internal variability, measured by the ensemble variance, over time and scenarios.** Each subplot presents historical time series (black) followed by the three scenarios SSP1-2.6 (blue), SSP2-4.5 (orange), or SSP5-8.5 (red) for a given model. The values written in each subplots correspond to 50-year averaged period of internal variability: 1900-1950 for historical (black) and 2050-2100 for scenarios (colors). A 30-year rolling averaged is applied to isolate long term trends.

### 3.3 Evolution of inter-model variability and uncertainty

Overall, the dominant factor in model-associated variability lies in the model main effect ($M$, Fig. 7). It follows the same two-step increase, described at the beginning with the statistical separation method (Fig. 2). As a reminder the two steps are associated with two uncertainties of AMOC response among models: (i) the response to the increase of aerosol concentration, and (ii) the magnitude of the AMOC decline. The low level of inter-model variability at the beginning of the simulations is artificially caused by the AMOC intensity reference taken as the individual model average over the 1850-1900 time period. It is difficult to analyze the $M$ main effect at the end of the simulations considering the differences between small and large ensemble models. Indeed, while large ensemble models present a decreasing inter-model variability due to the convergence of relative AMOC intensity decline, small ensemble models present an exponential increase of inter-model variability, linked to a divergence in terms of relative AMOC intensity decline. Taking aside potential inherent characteristics of small ensembles (which would break model democracy ; Knutti, 2010) and the impact of internal variability on the model trends, we interpret this as being also associated with the particular combination of models in the "small ensemble" and "large ensemble" category, which happen to exhibit different AMOC sensitivities to forcing. These results are also sensitive to the reference chosen for computing AMOC anomalies, as discussed in the method section. Indeed, when considering the absolute AMOC intensity, we observe a convergence in AMOC intensity among the small ensemble models and a divergence among the large ensemble models after 2040 (Fig. B3).

The second key property of the inter-model variability is the fact that it closely follows the evolution of physical components, especially internal variability. Most of the interactions involving models and realizations follow the same path as the corresponding interaction without the model dimension (Fig. 3 and 7). For instance, the interaction between models, time, and ensemble ($MRT$) evolves like the interaction between time and ensemble ($RT$). This is the same for $MR$ and $R$ or $SMRT$ and $SRT$. This illustrates that a fraction of internal variability is present in the model dimension and that averaging across models removes a part of internal variability—consistent with the logic previously discussed for the scenario dimension (see Section 2.2.2). This behavior reflects the fact that inter-model differences also manifest in terms of internal variability, which recent studies have linked to salinity biases and differences in dense water formation in the Labrador Sea, as well as to model horizontal resolution (Jackson et al., 2020, 2023).

In small ensemble models, $MT$ present a substantial level of variability prior to the 21$^{\text{st}}$ century, which indicates that small ensemble models were presenting different time evolution across this period and that a part of time variability has been removed with inter-model averaging. These greater inter-model differences in terms of trends is consistent with the greater AMOC sensitivity observed in models of the "small ensemble" category (Fig. 2).

### 3.4 Physical attribution of the ANOVA components

To summarize the previous findings on the ANOVA components, we propose a third approach—complementary to statistical separation and variance reconstruction—that organizes the variance components following a physical attribution. This physical attribution method distributes the terms into four different sources of variance (Fig. 8), following the interpretation of the

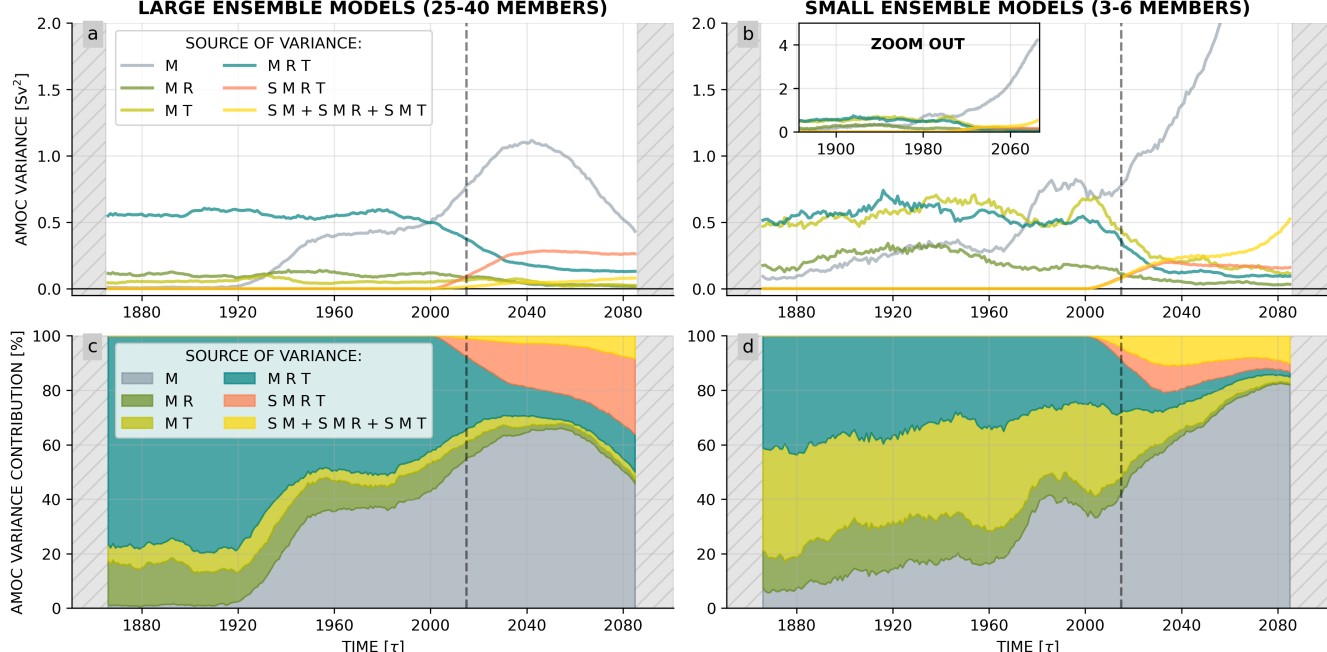

**Figure 7. Evolution of the model-associated factors of variability.** (a, b) Variance associated to model main effect and interactions. Small and large ensemble models have the same y-axis, and a zoom-out box is displayed for the variance of the small ensemble models to show the full increase in model contribution (b). (c, d) Variance contribution of model main effect and interactions to the total model-associated variability. Letters refer to the dimension involved in the variance calculation, with a single letter for main effects and a combination of letters for interactions: S refers to the scenario dimension, T to time, M to model, and R to realizations (see section 2.2.1). Results are presented for large ensemble models (a, c) with 25-40 members and small ensemble models (b, d) with 3-6 members.

ANOVA components detailed in section 2.2.2. The first source is the internal variability that corresponds to the mean ensemble variance (10) and gathers all term involving ensemble dimension ($R$, $RT$, $SR$, $MR$, $SRT$, $SMR$, $MRT$ and $SMRT$). This allows the fraction of internal variability located in each dimension to be robustly separated and analyzed as a distinct physical phenomena. The second source is the scenario-forced variability that aims at representing the effects of future forcing applied to the system and the AMOC uncertainty associated with future anthropogenic emissions. It is constructed from the scenario main effect and all interactions smoothing the scenario differences (associated with time and/or inter-model averaging: $S$, $ST$, $SM$, $SMT$). The third factor represents the dynamical adjustment associated with the response to the historical forcing located in the time main effect and the interaction between time and model that represents inter-model differences smoothing this response ($T$ and $MT$). Finally, the inter-model variability represents the evolution of inter-model differences over time and is only constituted by the model main effect ($M$).

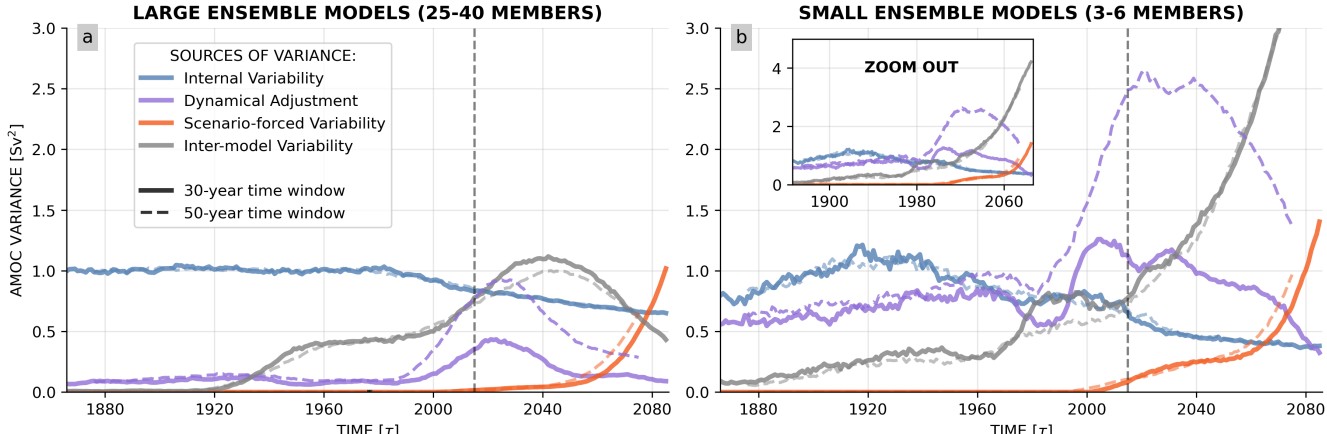

**Figure 8. Summary picture of the sources of AMOC variability.** (a, b) AMOC variance associated with each of the four sources of variance based on the physical attribution methodology described in section 4. Line style represents the size of the time window, either 30 (solid line), or 50 years (dashed line). Small and large ensemble models have the same y-axis, and a zoom-out box is displayed for the variance of the small ensemble models to show the full increase in model contribution. Results are presented for large ensemble models (a) with 25-40 members and small ensemble models (b) with 3-6 members.

## 4   Summary and discussion

The analysis of AMOC variability at 26°N in CMIP6 ensembles based on a 4-way analysis of variance, depicts three successive phases associated with three distinct regimes of AMOC variability, each dominated by a particular factor, that can be summarized using the physical attribution method presented previously.

1. The first phase from 1850 to around 1990 presents a relatively stable AMOC, whose variability is driven by internal variability, associated with time and ensemble dimension given the quasi-ergodicity of this regime;

2. The second phase from 1990 to 2050 is characterized by a decline of the AMOC associated with an adjustment to past historical emissions. It is thus a regime where time changes dominate, the forcing-scenario effect remaining weak. The importance of this regime is particularly evident when employing a 50-yr window for the ANOVA that allows to take better account of this declining trend (Fig. 8);

3. Finally the third phase starting around 2050 is a regime forced by emission scenarios, where the impact of anthropogenic
emissions pathways takes control on the divergence of the simulated trajectories.

An important finding of this work is the existence of two increasing phases of scenario variability. The first one, after 2015, rapidly stabilizes in a few decades. The second one, emerging after 2050, presents an exponential increase of the variability. During the first phase, scenarios simply act as little perturbations creating, in a way, "larger ensembles". This is particularly evident as the main signal associated with scenarios in this period lies in the interaction between scenarios, ensemble, and

time dimensions ($SRT$). This also leads us to detect an intermediate regime around 2040-2050, where most of the variability components stabilize and where the system is driven by this "scenario-perturbed" internal variability. In the second phase, the increase of variability is purely forced by the divergence of scenarios, and scenarios main effect takes the lead.

     The AMOC intensity decline associated with the transient increase of time variability, could suggest that this evolution is not forced by scenarios nor synchronous to anthropogenic emissions, because of the absence of clear scenario separation
before 2050. In this respect, AMOC behavior differs from global $CO_2$ concentration, anthropogenic radiative forcing, and temperature change, that already start diverging one or two decades earlier (O'Neill et al., 2016). Although this trend could be influenced by regional aerosol forcing (e.g., Deser et al., 2020; Menary et al., 2020; Robson et al., 2022), analyses of the local radiative imbalance over the North Atlantic (10°-60°N, 90°W-0°E) also show an early and rapid separation among scenarios (not shown).

This study also provides robust evidence that the AMOC internal variability in CMIP6 decreases in the projection period compared to historical reference. We also detected a potential link between this decline and emission pathways intensity. This is consistent with MacMartin et al. (2016) results with a single Earth System Model or Cheng et al. (2016) on interdecadal variability of the previous CMIP exercise. This also aligns with previous results on large-scale climate showing that internal variability is highly sensitive to forced variability and mean state (e.g., Coquereau et al., 2024). The decline of internal vari-
ability observed in the present work corresponds to a contraction of the phase-space and therefore an increasing predictability of the AMOC in the future from one time step to another. However, it does not substantially increase our ability to predict the future of the AMOC, since internal variability is a small component of the AMOC uncertainty, smaller than model or scenario-uncertainty after mid-century. Yet, internal variability seems to decrease concomitantly to the AMOC magnitude, thus it could be a potential proxy or warning signal of an AMOC decline, often hardly detectable because of the large AMOC interannual
variability (Lobelle et al., 2020). Furthermore, as seen in the results section, this progressive decline of internal variability coincide with the relocation of variance from $R$ and $RT$ to $SRT$. The physical attribution method facilitate the isolation of the internal variability decline signal without being impacted by the relocation of variance between different components related to internal variability.

     Finally, the physical attribution shows that model-only variability is someway over-estimated by the use of statistical sep-
aration and that model uncertainty, while being a major contributor, do not largely dominate the other factors. However, its contribution in AMOC projection uncertainty (mostly associated with AMOC absolute intensity and climate sensitivity, as discussed in Sec. 3.3 and in Weijer et al., 2020) has appeared as the strongest independent source of uncertainty (i.e. with the largest main effect), suggesting that further progress in AMOC modeling is needed to obtain more reliable AMOC projections.The disagreement between small and large ensemble models at the end of the 21[st] century raises interesting questions
about ensemble strategies and (i) the use of a small number of large ensemble models, versus (ii) the use of a large number of small ensemble models. There is thus a choice to be made between relying on few models (i) or not having a good separation of internal variability (ii). Here, all results are displayed separately between small and large ensemble models to give readers all the information. The fact that interactions involving model and ensemble dimensions follow the same evolution as the corresponding interactions without model dimension is interesting from an uncertainty perspective. Indeed, it underlines that

when the internal variability—which represents the range of possible states the system can occupy under a given forcing—is large, it is more difficult for the models to accurately find the correct state. Furthermore, the observed decline of ensemble variance when the AMOC intensity decreases, corresponds to a contraction of the space of possible states, and consequently one expects the absolute difference between models concerning this phase-space location to reduce.

A limitation of this study, tied to modeling capacity, lies in the relatively small number of large ensemble climate models and the limited size of most ensembles. This constraint is evident, for example, in the analysis of ensemble variance within individual models (Fig. 6), where variability exhibits noisy behavior and notable differences across small ensemble models, despite a generally consistent pattern of decline. Additionally, while some high-resolution climate simulations for a single member are becoming available, large-scale ensemble simulations remain restricted to relatively coarse spatial resolutions—on the order of one degree in the ocean. In this context, different research groups have highlighted the importance of fine-scale processes, such as mesoscale eddies, overflows and convection, in influencing the AMOC, raising concerns about the reliability of AMOC trends in coarse-resolution (one-degree) models (e.g., Hirschi et al., 2020; Jackson et al., 2020; Hewitt et al., 2022; Jackson et al., 2023; Gou et al., 2024). However, regarding AMOC strength at 26°N, a recent study found strong agreement between high- and low-resolution simulations (with 0.1° resolution in the ocean, Gou et al., 2024), which is the central focus of this work.

The ANOVA method employed in this study enables a detailed analysis of interaction terms between sources of variance, which is not possible with the widely used approach of Hawkins and Sutton (2009). This approach allows us to go beyond the assumption of additivity of variance associated with each individual dimension and directly examine cross-terms and interdependencies between different dimensions. With the ANOVA, by combining the main effect of each dimension with their interactions, we achieve a full reconstruction of the total variance. This contrasts with the previous method, where total variance was simply the sum of explicitly computed components.

*Code and data availability.* The program code (in Python) for computing the 4-way ANOVA is available at https://github.com/coquereau/ANOVA_4way. CMIP6 ensemble model outputs can be downloaded from the various Earth System Grid Federation (ESGF) nodes. The article references for each model can be found in Tab. 1.

*Author contributions.* Conceptualization: A.C., F.S., and Q.J. Methodology: A.C., F.S., T.H., J.J.H, and Q.J. Investigation: A.C., F.S., and Q.J. Visualization: A.C. Supervision: F.S., T.H., and J.J.H. Writing—original draft: A.C., F.S., and Q.J. Writing—review and editing: A.C., F.S., T.H., J.J.H, and Q.J.

*Competing interests.* The authors declare that they have no conflict of interest.

*Acknowledgements.* This work was supported by the ARVOR project funded by the LEFE IMAGO program, by the OceaniX project funded by the French ANR program, and by the ISblue project (Interdisciplinary graduate school for the blue planet, ANR-17-EURE-0015) funded by a grant from the French government under the program "Investissements d'Avenir". This work was also supported by the EERIE project (Grant Agreement No 101081383) funded by the European Union. Views and opinions expressed are however those of the authors only and do not necessarily reflect those of the European Union or the European Climate Infrastructure and Environment Executive Agency (CINEA). Neither the European Union nor the granting authority can be held responsible for them.

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

## Appendix A:  A stochastic model to interpret ANOVA interactions

To better understand the role and physical meaning of these interactions, we designed a minimalist synthetic model representing AMOC trajectories for different members and scenarios, with prescribed slopes and a superimposed random internal variability (see Fig. A1, first panel). The model is based on a stochastic mean-reverting process called Ornstein-Uhlenbeck process, similar to the model proposed in Hasselmann (1976). The evolution of our model is governed by the following equation:

$$d\psi_t = -\lambda(\psi_t - F_t)dt + \sigma(\psi_t)dW_t, \tag{A1}$$

with $\psi_t$ the AMOC intensity, $\lambda$ a damping term (also known as mean-reversion rate term, here equal to 0.3 yr$^{-1}$). In this model, we used a slight variation of a classical Ornstein-Uhlenbeck process (as used in Hasselmann, 1976) since the system is not pull toward the mean, but toward $F_t$ a given intensity that evolves over time. This term represents the forcing and drives the intensity decline. It is constant up to year 150 (i.e. $F_t = \psi_0 = 1$ Sv) and then declines, following a common trend across scenarios up to year 200 (i.e. $F_t = \psi_{150} + \beta t$, with $\beta = -0.006$ Sv yr$^{-1}$), and with a scenario-dependent trend afterward (i.e. $F_t = \psi_{200} + \gamma t$, with $\gamma$ ranging from $-0.01$ to $+0.002$ Sv yr$^{-1}$). $W_t$ is the Wiener process responsible for the stochastic fluctuations representing internal variability. The intensity of these fluctuations is set by $\sigma$ that depends on the AMOC intensity such that $\sigma(\psi_t) = \alpha\psi_t$, with $\alpha$ a constant, here equal to 0.04 yr$^{-1/2}$. Parameters were chosen to mimic the evolution of the AMOC time series observed in CMIP6 models. 1 000 members have been computed to ensure a robust evaluation of the internal variability and a 250-year spin-up has been performed to let the system adjust before the analysis.

## Appendix B:  Additional figures

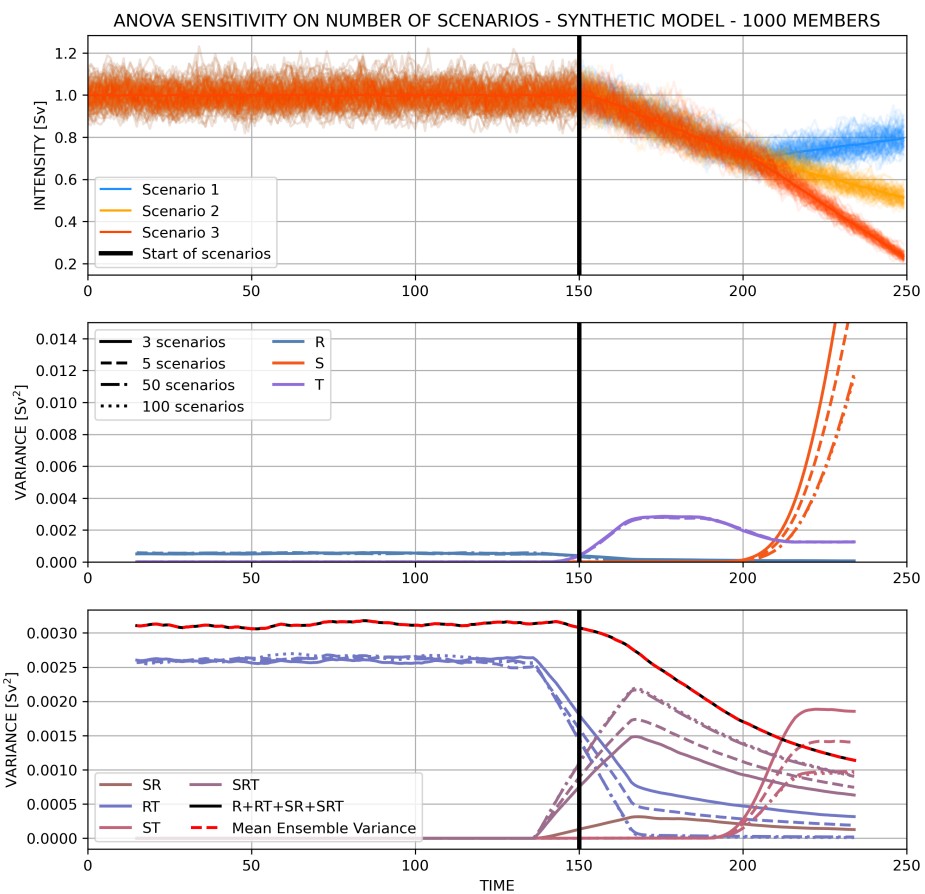

**Figure A1. ANOVA decomposition of the synthetic data set.** (Top panel) Synthetic AMOC trajectories for 1000 members and 3 scenarios, with a constant first period (years 0-150) where all scenarios are merged, a second period (years 150-200) with the same slope between scenarios but realizations show phase shifts in variability, and a third period (after year 200) where scenarios start to separate and follow different slopes. Internal variability is represented by a random Gaussian noise scaling with the AMOC intensity such that $\sigma(\psi_t) = \alpha\psi_t$, with $\alpha = 0.04$ yr$^{-1/2}$; $dt = 1$ yr. (Middle and bottom panels) ANOVA decomposition of the synthetic data set with separation between the main effects of time (T), scenarios (S), and realizations (R), as well as the interactions between these dimensions. The mean variance of the ensemble (red dotted line) is calculated as the variance of the ensemble (across realizations) averaged over the scenarios and over a sliding window of 30 years. The total internal variability from the ANOVA is calculated from the sum of all factors associated with the R dimension (R, RT, SR, and SRT, black line). The sensitivity of the results to the number of scenarios is represented by different lines of the same color: 3 scenarios in solid line, 5 scenarios in dashed line, 50 scenarios in dash-dotted line, and 100 scenarios in dotted line.

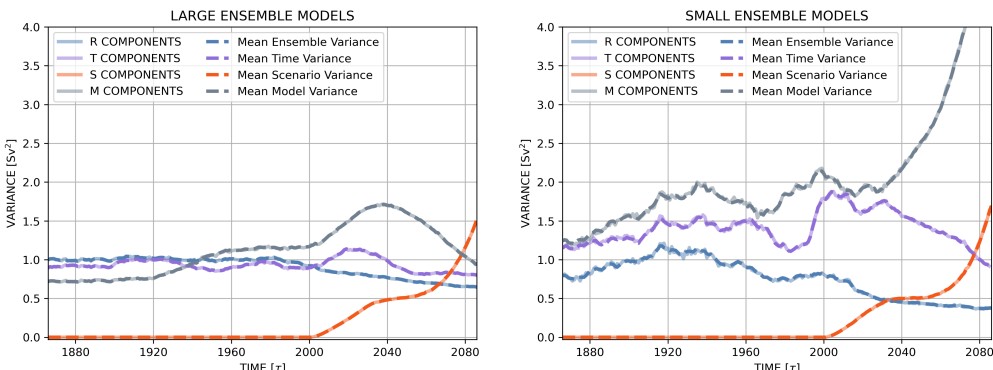

**Figure B1. Statistical variance reconstructed from ANOVA components.** The mean variance computed across each dimension is compared to the sum of all components involving this dimensions. Results are presented for large ensemble models (left) with 25-40 members and small ensemble models (right) with 3-6 members. Results indicate that the two methods lead to exactly equal time series.

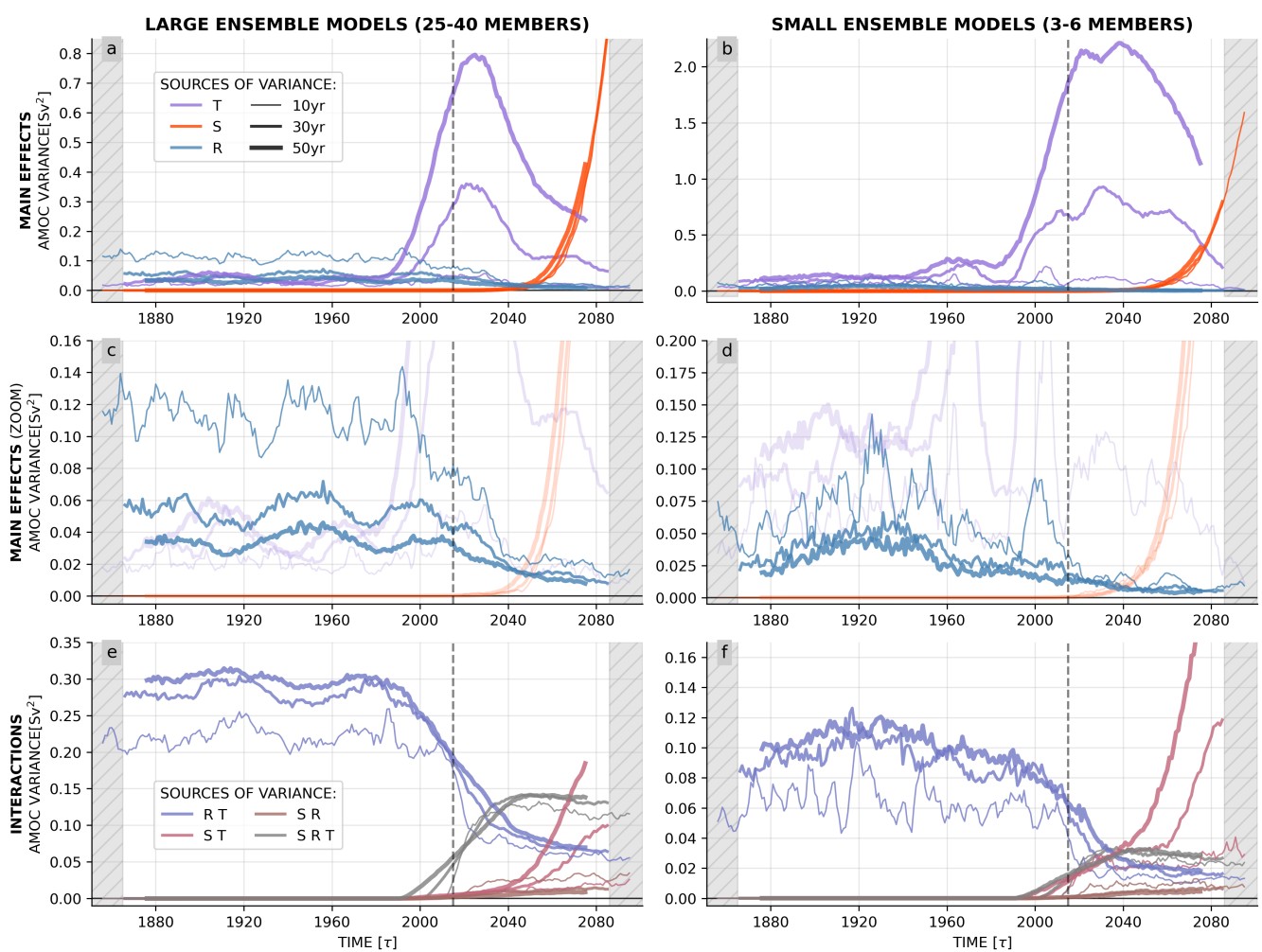

**Figure B2. Sensitivity test on time window size.** Variance associated to each main effect (a-d) and interaction (e,f). (c,d) Zoom on ensemble main effect. Line thickness represents the size of the time window from 10 (thin) to 50 years (thick). Results are presented for large ensemble models (a, c, e) with 25-40 members and small ensemble models (b, d, f) with 3-6 members.

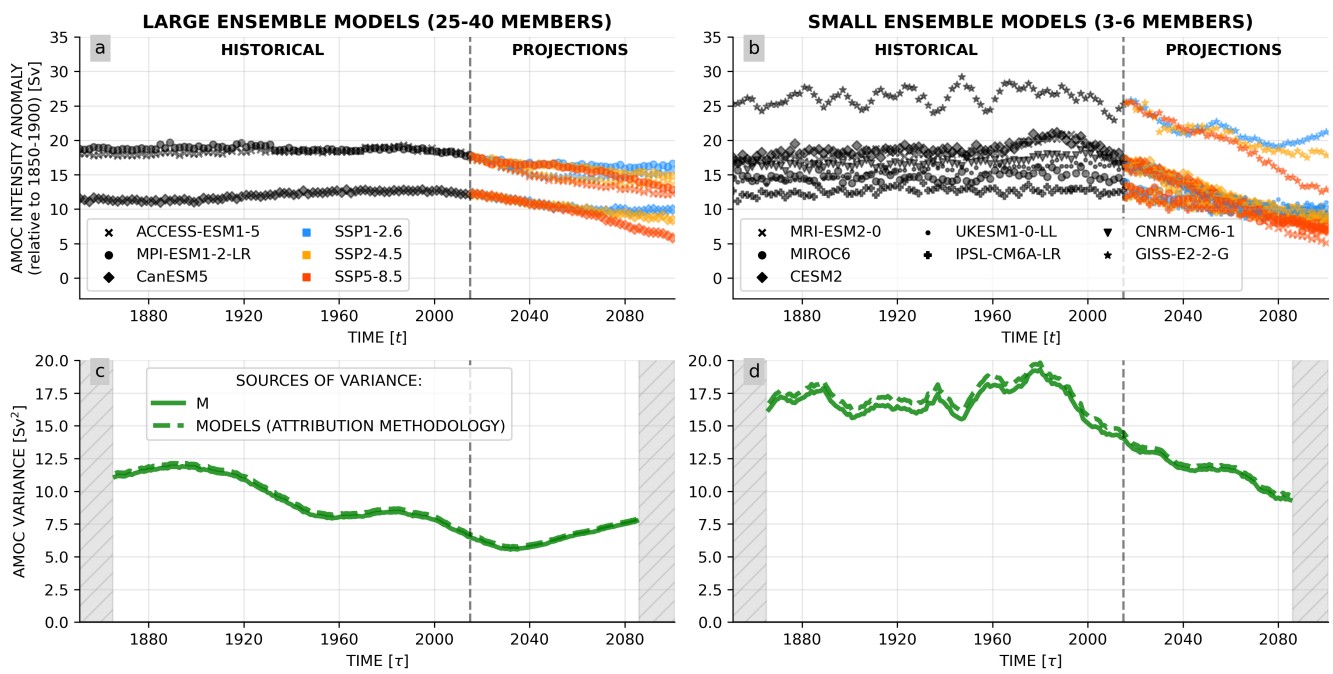

**Figure B3. Sensitivity test with AMOC absolute intensity.** (a, b) AMOC absolute intensity time series over historical (1850-2015, black) and projection (2015-2100, colors) periods. (c, d) Variance associated with model main effect only (solid line) and with all model-associated factors combined with the separation methodology by Zhang et al. (2023) (dashed line). Values correspond to absolute AMOC intensity. Results are presented for large ensemble models (a, c) with 25-40 members and small ensemble models (b, d) with 3-6 members.

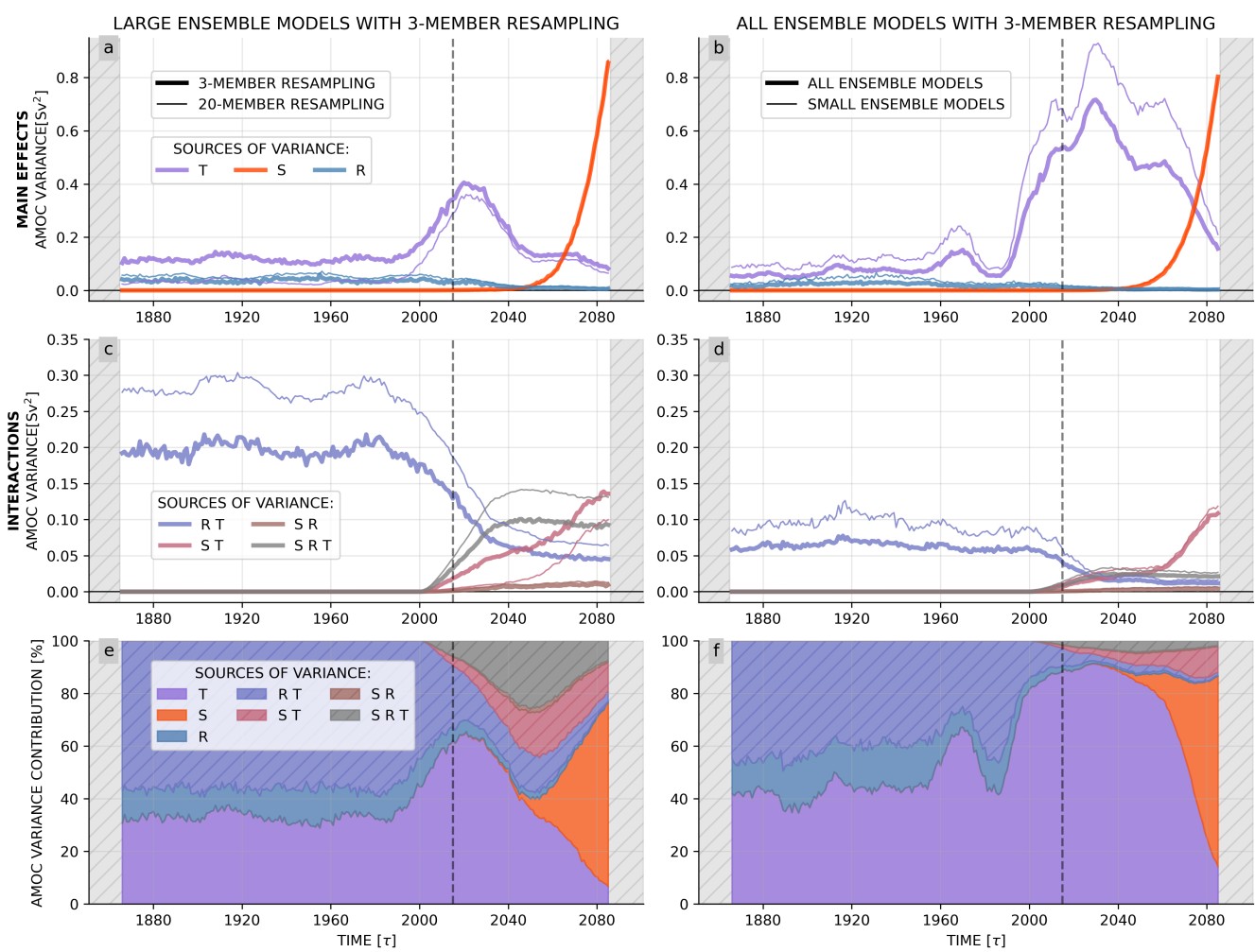

**Figure B4. Sensitivity test on bootstrapping methodology: application to the variability of physical factors.** (a-d) Variance associated to each main effect (a, b) and their interaction (c, d). (e, f) Relative variance contribution of each main effect and their interactions to the total physical factor variability. Left-hand panels show the sensitivity of the large ensemble results to the number of members in each resampling. Thin lines present the control experiment with 20 members of each model (identical to Fig. 3 a, c, e) and thick lines present the results obtained using 3 members of each model per resampling. Right-hand panels show the sensitivity to mixing large and small ensemble models. Thin lines present the control experiment with only small ensemble models (identical to Fig. 3 b, d, f) and thick lines present the results obtained with all ensembles. 3-member resampling are used here.

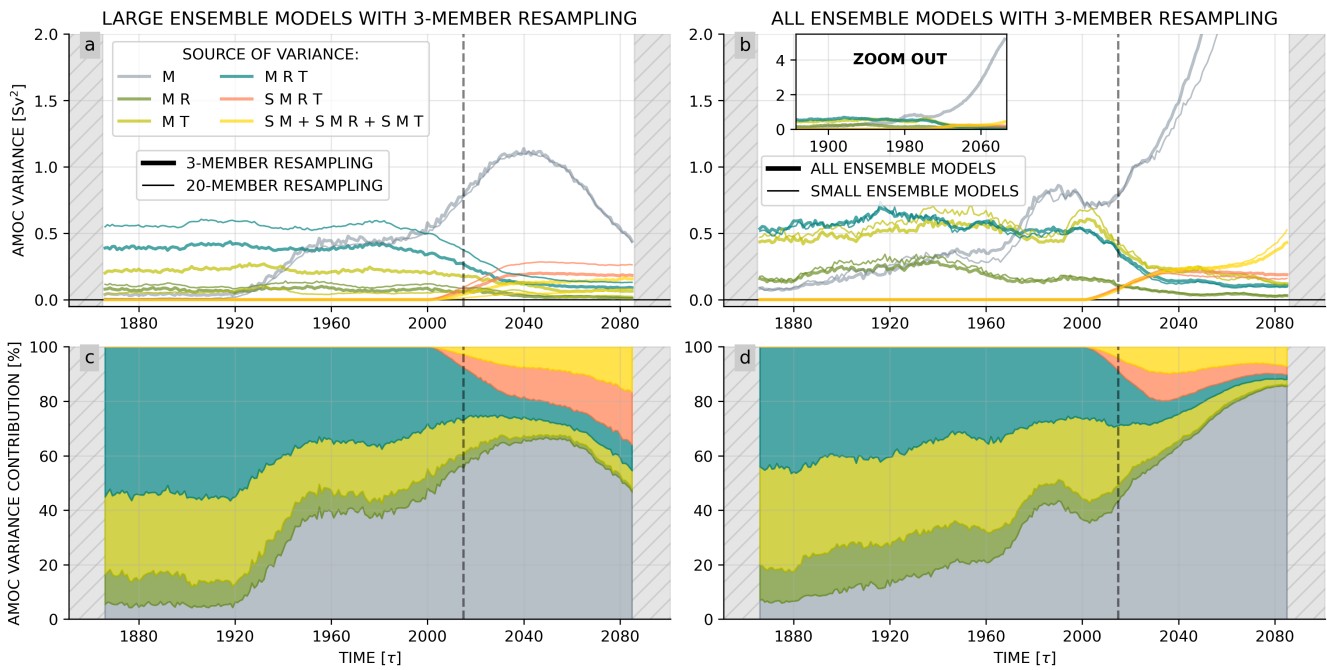

**Figure B5. Sensitivity test on bootstrapping methodology: application to the variability of model-associated factors** (a, b) Variance associated to model main effect and interactions. (c, d) Variance contribution of model main effect and interactions to the total model-associated variability. Left-hand panels show the sensitivity of the large ensemble results to the number of members in each resampling. Thin lines present the control experiment with 20 members of each model (identical to Fig. 3 a, c, e) and thick lines present the results obtained using 3 members of each model per resampling. Right-hand panels show the sensitivity to mixing large and small ensemble models. Thin lines present the control experiment with only small ensemble models (identical to Fig. 3 b, d, f) and thick lines present the results obtained with all ensembles. 3-member resampling are used here.

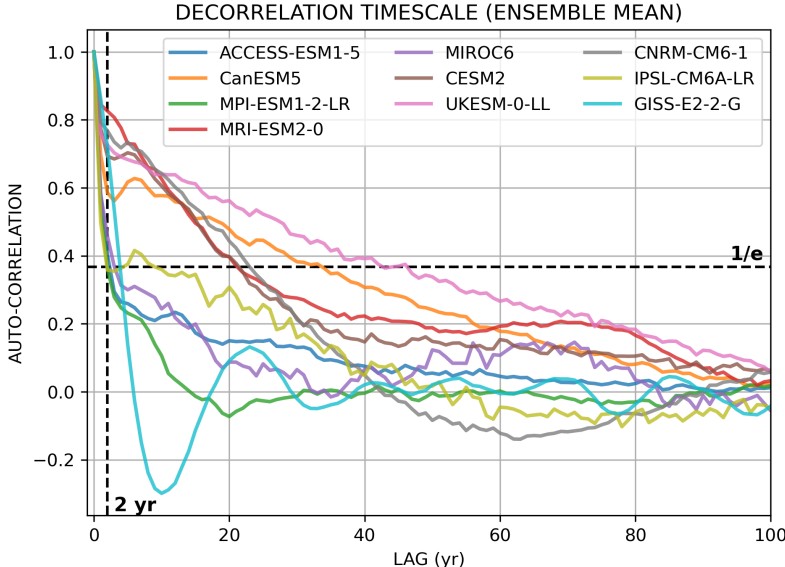

**Figure B6. Decorrelation timescale of AMOC intensity time series for each model.** Evolution of the normalized auto-correlation with respect to the considered lag. Decorrelation timescale corresponds to the the e-folding timescale, i.e. the lag when the normalized auto-correlation falls below $1/e$.