# Peer review of "Past, Present, and Future Variability of Atlantic Meridional Overturning Circulation in CMIP6 Ensembles"

_EGUsphere, 2025_

## Author Comment (AC2)

**Authors' response to Reviewer #2 comments on "Past, Present, and Future Variability of Atlantic Meridional Overturning Circulation in CMIP6 Ensembles"**

We would like to thank the reviewer for the helpful comments and suggestions. These will help us to improve our manuscript considerably, especially the comments on the interpretation of interactions which led to further work and helped us to better understand the physical significance of these interactions. In addition, this new interpretation has shown us that Fig. 4 in the first version of the manuscript should be based on main effects and interactions instead of just the main effects. The updated figure is presented at the end of this response and does not change the interpretation of the results.

Finally, we will respond to each reviewer's comment below and provide a revised version of the manuscript at the end of the interactive discussion.

Arthur Coquereau, Florian Sévellec, Thierry Huck, Joël J.-M. Hirschi, and Quentin Jamet

**Review**

1. I miss an interpretation of what I see in Fig. 1. The caption explains what each curve represents, but how to interpret the differences is unclear to me and not (well) discussed in the text.

We agree with the reviewer about the lack of interpretation. This figure is mainly qualitative, illustrating what is involved in averaging a dimension. In the various sub-plots, we can see the total variability of the dimensions studied and how this is reduced when averaging certain dimensions. For example, in Fig. 1a, $x_\tau(r, t)$ represents the total variance in the time and ensemble dimensions. Then, variability decreases

when averaging time, as observed in $x_\tau(r)$ or by averaging realizations in $x_\tau(t)$. These figures are intended to help understand the impact of the averaging process and to observe the dominant dimensions. We will add this information about the purpose of the figure in the revised manuscript.

2. In the same Fig. different y-axes are used, which is misleading. I would suggest to you use the same y-scale everywhere (the one from panel f), or at least the panel f scale in b,d,f and the panel e scale in a,c,e.

We agree with this comment and will update the figure using the same y-scale for a, c and e on the left side and b, d and f on the right side.

3. I think it is misleading to talk about the "large ensembles" and the "small ensembles" in many places, suggesting that differences in AMOC variability are due to ensemble size, while it is most probably to different models being analysed and more and less models available in the small and large ensemble. I suggest to use "The models with a large ensemble" vs "the models with a smaller ensemble", or just ensemble A and ensemble B. This especially holds to the amount of decline. The much larger decline in the small ensemble is simply due to including models with larger AMOC sensitivity while the large ensemble by chance consists of 3 models that do not weaken much. This fact has nothing to do with ensemble size. On the other hand, when discussing the interactions I accept that ensemble size can be driving factor for differences in certain periods.

We agree that some of the differences between small and large ensembles are not directly due to the size of the ensemble, but rather to the characteristics of the models, as suggested by the reviewer with the sensitivity of AMOC to forcing. This aspect of sensitivity is probably reflected in the factors and interactions associated with the time and scenario dimensions. The factors and interactions associated with scenarios behave relatively similarly between small and large ensembles. Qualitatively, this is also the case for the main effect of time, but not quantitatively. The greater amplitude of the time main effect (T) bump for small ensembles and its greater spread over time could be due to the sensitivity of AMOC to forcing. However, this may also be linked to the small size of the ensembles, which does not allow internal variability to be separated as precisely as for larger ensembles. The reviewer's suggestion also provides a convincing

explanation for the stronger model-time interaction (MT) in small ensembles, as well as for the model main effect (M). We will add these arguments in the manuscript. We note, however, that the separation between small and large ensembles remains justified in view of the factors associated with internal variability. Finally, we agree with the reviewer concerning the suggestion of defining clearly the two distinct categories of models. We will define and use the names "Large ensemble models" and "Small ensemble models" throughout the revised manuscript.

4. Please say what R T, S mean in the caption of Fig. 3. Figures should be stand-alone understandable without having to go back to the text to see what they actually display.

We agree with this comment and will add the explanation of the dimension letters in the caption, with R for realizations (ensemble members), T for time, and S for scenarios.

5. Showing the interactions is of interest but I completely miss a physical picture of how I should interpret these interactions and what physical process they represent. What does it mean? Please explain in the text.

As mentioned in section 2.2.1, "interaction occurs when the separate effects of the factors do not combine additively". They thus represent the fraction of variance that is inherently located in two or more dimensions. We agree with the reviewer's point that a physical interpretation of these interactions is missing in the manuscript. Further investigations have helped us clarify the physical role of these components. We show that the interactions between "physical" factors involving ensemble dimension represents internal variability (i.e.. SR, RT, SRT). The interaction between scenarios and time (ST) represents the fraction of scenario variability smoothed by the rolling time window. Below, we provide further elements to justify the physical interpretation and understanding of the interactions. We will add these elements in the revised manuscript.

To better understand the role and physical meaning of these interactions, we designed a minimalist synthetic model representing AMOC trajectories for different members and scenarios, with prescribed slopes and a superimposed random internal variability (see Fig. R1, first panel). The model is based on a stochastic mean-reverting process called

Ornstein-Uhlenbeck process, similar to the classical model proposed in Frankignoul and Hasselmann (1977). The evolution of our model is governed by the following equation:

$$d\psi_t = -\lambda(\psi_t - F_t)dt + \sigma(\psi_t)dW_t ,$$

with $\psi_t$ the AMOC intensity, $\lambda$ a damping term (also known as mean-reversion rate term, here equal to $0.3\ yr^{-1}$). Here, we used a slight variation of the classical model used in Frankignoul and Hasselmann (1977) since the system is not pull toward the mean, but toward $F$ a given intensity that evolves over time. This term represents the forcing and drives the intensity decline. It is constant up to year 150 (i.e., $F_t = \psi_0 = 1\ Sv$) and then declines, following a common trend across scenarios up to year 200 (i.e., $F_t = \psi_{150} - \beta t$, with $\beta = -0.006\ Sv.yr^{-1}$), and with a scenario-dependent trend afterward (i.e., $F_t = \psi_{200} - \gamma t$, with $\gamma$ the slope associated with a given scenario, here, ranging from $-0.01$ to $+0.002\ Sv.yr^{-1}$). $W_t$ is the Wiener process responsible for the stochastic fluctuations representing internal variability. The intensity of these fluctuations is set by $\sigma$ that depends on the AMOC intensity such that $\sigma(\psi_t) = \alpha\psi_t$, with $\alpha$ a constant. Parameters were chosen to mimic the evolution of the AMOC time series observed in CMIP6 models. 1000 members have been computed to ensure a robust evaluation of the internal variability and a 250-year spin-up has been performed to let the system adjust before the analysis.

The variability of the synthetic dataset constructed with this model has then been decomposed using the ANOVA method. This approach helps us understand the sensitivity to different parameters, particularly the number of scenarios, which is challenging to test with "real" AMOC trajectories from climate models. Here, we focus on physical interactions, excluding model dimensions, as model-associated interactions primarily represent model uncertainty associated with interactions.

Approaching these interactions from the perspective of Internal Variability (IV) makes it easier to understand their physical significance. First we compute the mean ensemble variance of the synthetic dataset as a reference, since it is often used as a proxy for IV (e.g., Coquereau et al., 2024). With this simple dataset, we observe that the mean ensemble variance is exactly equal to the sum of the ensemble main effect (R) and all interactions involving the ensemble dimensions (RT, SR, and SRT, without weighting). This highlights that each dimension includes a part of IV. For time and ensemble, this is evident, but it is also true for scenarios. When the trajectories separate under different

forcing scenarios, the phasing of their variability changes. This phase shift represents the fraction of IV associated with the scenarios. Averaging over scenarios removes part of this phase shift and, consequently, part of the IV, similar to averaging over a time window or realizations. If all dimensions are sufficiently large (many members, many scenarios, long time period), the SRT component should capture all the IV. However, if the dimensions are too small, SRT will not capture the entire IV, and the importance of SRT decreases, compensated by an increase in other components/interactions. This is, for instance, highlighted by the progressive decrease of SRT and increase of RT when using fewer scenarios. The same logic applies for time dimension. At the beginning, R represents the part of the internal variability that is not captured in RT, i.e., the internal variability with a period larger than the rolling time window of 30 yr. The sensitivity test presented in Fig. A1 (Supplementary Material of the manuscript), shows the decrease of R and increase of RT when increasing the time window and demonstrates that R represents the low-frequency IV with a period larger than the window. Thus, when the size of the time window increases there is, logically, less variability at lower frequency. After the beginning of scenarios this low-frequency internal variability shifts from R to SR.

The fact that all components involving S are null at the beginning of the time series is due to the fact that all scenarios are merged, making SR and SRT equal to zero because averaging over scenarios does not remove any IV.

Finally, ST represents the part of the scenario dispersion removed by the inherent smoothing of the time averaging, as shown in the sensitivity test on time window size (Fig. A1, Supplementary Material), where increasing the window size decreases S and increases ST, and vice versa.

To summarize:
**Total Internal Variability:** R+RT+SR+SRT (with equal weights)
**Low-Frequency Internal Variability:** R+RS (period larger than the time window)
**Fraction of Internal Variability Removed by Averaging:**
- **Over time and scenarios:** SRT
- **Over time only:** RT
- **Over scenarios only:** SR

**Internal Variability Included in Specific Dimensions:**
- **Time dimension:** RT+SRT
- **Scenario dimension:** SR+SRT

[Figure]

**Fig. R1 - ANOVA decomposition of the synthetic data set.** (Top panel) Synthetic AMOC trajectories for 1000 members and 3 scenarios, with a constant first period (years 0-150) where all scenarios are merged, a second period (years 150-200) with the same slope between scenarios but realizations show phase shifts in variability, and a third period (after year 200) where scenarios start to separate and follow different slopes. Internal variability is represented by a random Gaussian noise scaling with the AMOC intensity such that $\sigma(\psi_t) = \alpha\psi_t$, with $\alpha = 0.04$; dt = 1 yr. (Middle and bottom panels) ANOVA decomposition of the synthetic data set with separation between the main effects of time (T), scenarios (S) and realizations (R) and the interactions between these dimensions. The mean variance of the ensemble (red dotted line) is calculated as the variance of the ensemble (across realizations) averaged over the scenarios and

over a sliding window of 30 years. The total internal variability from the ANOVA is calculated from the sum of all factors associated with the R dimension (R, RT, SR and SRT, black line). The sensitivity of the results to the number of scenarios is represented by different lines of the same color: 3 scenarios in solid line, 5 scenarios in dashed line, 50 scenarios in dash-dotted line, and 100 scenarios in dotted line.

6. I would rephrase the conclusion between anthropogenic forcing intensity (which could be positive as well in terms of AMOC: aerosols!) and ensemble variance decrease or decrease of natural variability by saying there is a strong link between forced AMOC weakening and decrease in ….(l 318)

We agree with this suggestion and will update the text accordingly.

7. Discussion on line 319-320 and further down in 3.2.2. I see what you say in your figures, but don't have a clear picture of what SRT interaction actually means in terms of physical processes, so this part is not meaningful to me (see also point 5).

We agree with the reviewer that we lack a physical interpretation of the interactions. We have added physical explanations in the response to point 5 and will update the manuscript to include this information.

8. My physical interpretation of 2.3 is that up to 2050 scenarios with stronger greenhouse forcing have smaller weakening in aerosol emissions and vice versa, making the impact on AMOC almost scenario independent, until the aerosol effect is gone, and we see the effect of greenhouse gas forcing alone.

The reviewer seems to suggest that the delayed emergence of the main scenario effect on variability is associated with the compensating effect of greenhouse gas (GHG) emissions and aerosol forcing. The SSP1-2.6 scenario shows a smaller increase in GHGs and a larger decrease in aerosols, and vice versa for the SSP5-8.5 scenario. However, this is not consistent with the evolution of the net radiative forcing imbalance (which account for both GHG and aerosol), which shows a divergence of the scenarios rapidly after their start in 2015 (O'Neill et al., 2016).

9. Discussion around line 360. I disagree with the interpretation, see point 3.

We agree with the reviewer and will update the manuscript to explain that this difference between small and large ensembles regarding model main effect (M) is not associated with the size of the ensembles but with the individual characteristics of the models selected in each category.

10. Summary your point 2. I think what you see in phase 2 is also forced by SSP scenarios, but as said before the net forcing on AMOC is reasonably equal for the scenarios. SSP126 forces stronger AMOC weakening than SSP245 and SSP585 in the first half of this century because the aerosols are faster removed from the atmosphere in this scenario.

In accordance with our response to point 8, this interpretation does not seem compatible with the trajectories of net radiative forcing imbalance in CMIP6, which separates between scenarios rapidly after 2015.

**FIGURE 4 - UPDATED**

[Figure]

References:

Coquereau, A., F. Sévellec, T. Huck, J. J. Hirschi, and A. Hochet, 2024: Anthropogenic Changes in Interannual-to-Decadal Climate Variability in CMIP6 Multiensemble Simulations. J. Climate, 37, 3723–3739, https://doi.org/10.1175/JCLI-D-23-0606.1.

Frankignoul, C. and Hasselmann, K. (1977), Stochastic climate models, Part II Application to sea-surface temperature anomalies and thermocline variability. Tellus, 29: 289-305. https://doi.org/10.1111/j.2153-3490.1977.tb00740.x

O'Neill, B. C., Tebaldi, C., van Vuuren, D. P., Eyring, V., Friedlingstein, P., Hurtt, G., Knutti, R., Kriegler, E., Lamarque, J.-F., Lowe, J., Meehl, G. A., Moss, R., Riahi, K., and

Sanderson, B. M.: The Scenario Model Intercomparison Project (ScenarioMIP) for CMIP6, Geosci. Model Dev., 9, 3461–3482, https://doi.org/10.5194/gmd-9-3461-2016, 2016.